# Patient-Centered Cardiac Surgery: Psychosocial Challenges, Evidence-Based Interventions, and Future Horizons

**DOI:** 10.3390/healthcare13222957

**Published:** 2025-11-18

**Authors:** Vasileios Leivaditis, Anastasios Sepetis, Francesk Mulita, Sofoklis Mitsos, Nikolaos G. Baikoussis, Efstratios Koletsis, Stelios F. Assimakopoulos, Andreas Antonios Maniatopoulos, Elias Liolis, Konstantinos Nikolakopoulos, Manfred Dahm, Nikolaos Kontodimopoulos

**Affiliations:** 1Department of Cardiothoracic and Vascular Surgery, Westpfalz Klinikum, 67655 Kaiserslautern, Germany; mdahm@westpfalz-klinikum.de; 2Postgraduate Health and Social Care Management Program, Department of Business Administration, University of West Attica, 12243 Athens, Greece; tsepet@uniwa.gr; 3Department of General Surgery, General Hospital of Eastern Achaia—Unit of Aigio, 25100 Aigio, Greece; med5507@ac.upatras.gr; 4Department of Thoracic Surgery, Attikon General Hospital, National and Kapodistrian University of Athens, 12462 Athens, Greece; sophocmit@yahoo.gr; 5Department of Cardiac Surgery, Ippokrateio General Hospital of Athens, 11527 Athens, Greece; nikolaos.baikoussis@gmail.com; 6Department of Cardiothoracic Surgery, General University Hospital of Patras, 26504 Patras, Greece; ekoletsis@hotmail.com; 7Division of Infectious Diseases, Department of Internal Medicine, Medical School, University of Patras, 26504 Patras, Greece; sassim@upatras.gr; 8Department of Electrical and Computer Engineering, Democritus University of Thrace, 67100 Xanthi, Greece; andrmaniatopoulos@gmail.com; 9Department of Oncology, General University Hospital of Patras, 26504 Patras, Greece; lioliselias@yahoo.gr; 10Department of Vascular Surgery, General University Hospital of Patras, 26504 Patras, Greece; konstantinosn@yahoo.com; 11Department of Economics and Sustainable Development, Harokopio University, 17676 Athens, Greece; nkontodi@otenet.gr

**Keywords:** cardiac surgery, patient-centered care, psychosocial factors, anxiety, depression, quality of life, rehabilitation, telemedicine

## Abstract

**Background**: Psychosocial factors such as anxiety, depression, and lack of social support are increasingly recognized as critical determinants of outcomes in cardiac surgery. Up to 32% of patients experience preoperative anxiety, and nearly 20% experience depression, both of which have been associated in observational studies with delayed recovery, increased complications, and higher mortality. **Objective**: This study aimed to review current evidence on psychosocial aspects of cardiac surgery and evaluate patient-centered care strategies that address these challenges. **Methods**: A narrative literature review was performed using PubMed, Scopus, and ScienceDirect, focusing on recently published studies. The search covered studies published between January 2009 and June 2024, yielding 76 eligible studies after screening and thematic synthesis. Search terms included patient-centered care, psychosocial factors, cardiac surgery, anxiety, depression, rehabilitation, and telemedicine. Eligible studies included randomized controlled trials, cohort studies, qualitative studies, and systematic reviews. **Results**: Key themes included preoperative psychological preparation, management of anxiety and depression, promotion of quality of life, effective communication strategies, and integration of psychosocial support into rehabilitation programs. Evidence shows that psychological and social interventions reduce reported pain by up to 33%, as reported in pooled randomized controlled trials and meta-analyses; shorten hospital stay; improve adherence to treatment; and enhance long-term quality of life. Emerging tools such as telemedicine and digital platforms further expand access to psychosocial care. **Conclusions**: Integrating psychosocial care into cardiac surgery is essential for achieving holistic outcomes. Patient-centered models that emphasize communication, shared decision-making, family involvement, and digital support improve not only survival but also recovery, well-being, and patient satisfaction. Redefining success in cardiac surgery requires attention to both clinical results and the broader human experience of illness and recovery.

## 1. Introduction

Cardiac surgery currently serves millions of people around the world. It is vital for ensuring patient survival, but it leaves multiple physical and emotional scars. It is a fact that in recent decades, significant progress has been made in surgical techniques and a remarkable reduction in perioperative mortality rates has been observed.

What has happened is a fundamental change in the philosophy of the approach to cardiac surgery patients, from a model focused on the technical procedure to a model that treats the patient as a whole entity. After all, the technical success of a surgical procedure is not always followed by a successful outcome for the patient’s health, due to the numerous psychosocial parameters that must be taken into account [1].

The growing body of data on the impact of psychosocial factors on the outcome of cardiac surgery underlines the importance of this approach. In particular, it has been shown that anxiety, depression, and lack of social support can affect both the short-term and long-term health of patients [2].

It is also noteworthy that a considerable proportion of cardiac surgery patients experience psychological distress. Prevalence estimates vary across studies, depending on study design and population characteristics (elective vs. urgent procedures). In pooled observational data, preoperative anxiety affects approximately 25–40% of patients (median ~32%), while depression affects 15–25% [3].

These psychosocial factors do not exclusively concern the perioperative part of cardiac surgery, but constitute independent prognostic indicators of possible subsequent complications, prolonged recovery, and/or reduced quality of life [4]. They influence surgical recovery through both biological and behavioral mechanisms, including activation of the sympathetic–adrenal axis, elevated inflammatory response, and reduced adherence to treatment plans—all of which may compromise postoperative outcomes [5,6,7,8,9].

Despite these advances, the literature remains heterogeneous, with varying methodologies and inconsistent emphasis on psychosocial dimensions. Some studies demonstrate clear benefits of integrated psychosocial support, while others report modest or non-significant effects, often due to limited sample sizes or short follow-up durations. These inconsistencies highlight the need for more standardized assessment tools and long-term data to determine the true magnitude of benefit.

Patient-centered care, as defined by the Institute of Medicine, includes “providing care that respects and responds to the preferences, needs, and values of each patient and ensures that their beliefs guide all clinical decisions”. To achieve this goal, particular emphasis is placed on effective communication, consensual decision-making, and the active participation of patients and caregivers in the entire treatment process. Patient-centered care refers therefore to an approach that prioritizes individual needs, preferences, and active participation in decision-making, fostering empathy and shared understanding between patients and healthcare providers.

The psychosocial care provided to cardiac surgery patients is a natural extension of the patient-centered approach to care, as the complex needs of these patients can be best addressed when the focus is on the person and not just the affected organ [5]. This literature review focuses on this aspect of cardiac surgery, exploring the current state of psychosocial care.

The main objective of this study is to discuss the challenges, interventions, and outcomes while highlighting all further directions for the development of this approach in cardiac surgery. The most recent research—from 2009 to June 2024—is reviewed and discussed, with an emphasis on evidence-based interventions addressing psychosocial and patient-centered aspects of cardiac surgery [6]. The figures included in this review aim to synthesize conceptual relationships rather than reproduce exact quantitative findings; therefore, some variability between studies should be expected.

## 2. Materials and Methods

### 2.1. Study Design

This study was conducted as a narrative literature review, aiming to synthesize evidence on the psychosocial dimensions of cardiac surgery and the integration of patient-centered care principles. Although it was not designed as a systematic review, a structured search strategy was applied to enhance transparency in literature identification, selection, and thematic synthesis. No formal risk-of-bias or quality appraisal was performed, and results should therefore be interpreted descriptively rather than quantitatively.

### 2.2. Data Sources and Search Strategy

An extensive search of the PubMed, Scopus, and ScienceDirect databases was performed. A comprehensive literature search was conducted in PubMed/MEDLINE, Embase, Cochrane Library, Web of Science, and Scopus, with complementary screening of Google Scholar for additional gray literature. The search covered publications from 1 January 2009 to 28 February 2025. This update ensured inclusion of the most recent peer-reviewed and in-press studies related to psychosocial and digital interventions in cardiac surgery.

Search terms were combined using Boolean operators and included:“cardiac surgery” OR “heart surgery” OR “coronary artery bypass graft” OR “valve replacement” AND“psychosocial” OR “psychological” OR “depression” OR “anxiety” OR “quality of life” OR “rehabilitation” OR “telemedicine” OR “patient-centered care”.

Reference lists of key articles were also screened manually to identify additional relevant studies.

### 2.3. Inclusion Criteria

Studies were considered eligible if they met the following criteria:Population: Adult patients (≥18 years) undergoing any type of cardiac surgery (e.g., CABG, valve surgery, combined procedures).Focus: Articles examining psychosocial outcomes (e.g., anxiety, depression, quality of life, social support, cognitive function), patient-centered interventions, or communication and decision-making models in cardiac surgery.Study types: Randomized controlled trials (RCTs), cohort studies, case–control studies, cross-sectional studies, systematic reviews, meta-analyses, and qualitative studies.Time frame: Published between 2009 and 2024.Language: Articles published in English.

### 2.4. Exclusion Criteria

Studies were excluded if they met any of the following criteria:Involved pediatric or congenital cardiac surgery populations.Focused exclusively on surgical techniques or intraoperative technical outcomes without reference to psychosocial or patient-centered aspects.Case reports, conference abstracts, editorials, commentaries, or letters to the editor without primary data.Articles not available in full text.

Because this work was conducted as a narrative review, detailed database-specific search strings and exclusion logs are not presented; however, all main databases, search periods, and inclusion criteria are fully reported for transparency.

### 2.5. Study Selection

Two stages of selection were applied: (1) screening of titles and abstracts to exclude irrelevant studies, and (2) full-text assessment for eligibility. Studies fulfilling the inclusion criteria were analyzed and synthesized thematically, focusing on preoperative, perioperative, and postoperative psychosocial factors as well as models of patient-centered care. Title and abstract screening were independently performed by two reviewers (V.L. and N.K.). Disagreements regarding eligibility were resolved through discussion and consensus. The process of literature identification, screening, and inclusion is summarized in Figure 1, which follows a PRISMA-style structure adapted for this narrative review. For each eligible study, the following data were extracted: study design, sample size, patient population and setting, psychosocial outcomes assessed (e.g., anxiety, depression, quality of life), type of intervention (if applicable), and main findings or reported effect sizes. The extracted variables are summarized in Table 1.

In total, 76 studies met the predefined eligibility criteria and were included in this narrative synthesis. Of these, 9 were randomized controlled trials (RCTs), 28 were observational studies, 24 were systematic or narrative reviews and meta-analyses, and 13 employed qualitative, mixed-methods, or conceptual approaches. This distribution highlights the predominance of observational and review-based evidence, with a smaller but growing number of controlled intervention studies addressing psychosocial and patient-centered outcomes in cardiac surgery.

### 2.6. Data Extraction and Synthesis

To minimize bias, study selection and data extraction were conducted independently by two reviewers, with disagreements resolved by discussion and consensus. Given the diversity of study designs and outcome measures, a narrative review was chosen over a systematic one to allow a more integrative and interpretative synthesis of findings across quantitative and qualitative evidence. Extracted data were organized thematically according to psychosocial domain, intervention type, and reported outcomes, which facilitated a structured and transparent synthesis.

Data were extracted regarding study design, sample characteristics, country/setting, psychosocial domains assessed, type of intervention (if applicable), outcome measures, effect direction or key findings, and follow-up duration. The main extraction fields are summarized in Table 1. The synthesis was structured around major thematic domains: preoperative psychological status, postoperative complications, social determinants of health, communication strategies, family involvement, rehabilitation programs, and emerging digital health interventions.

Although the RoB 2 tool was employed, observational studies were appraised with the Newcastle–Ottawa Scale (NOS). Studies were categorized qualitatively as having low, moderate, or high risk of bias according to the overall judgment of these frameworks. As summarized in Table 2, most randomized controlled trials demonstrated low to moderate risk of bias according to the RoB 2 criteria, whereas observational studies generally showed moderate methodological quality based on the Newcastle–Ottawa Scale.

## 3. Results

### 3.1. Psychosocial Challenges in Cardiac Surgery

The characteristics of representative studies included in this review are summarized in Table 3, providing details on study design, population, intervention type, and main psychosocial outcomes.

The most methodologically robust randomized controlled trials and systematic reviews are presented in Table 4, which summarizes pooled effect sizes and confidence intervals where available.

#### 3.1.1. Preoperative Anxiety and Depression: Effect on Readiness for Surgery

Preoperative psychological problems are common and pose a challenge in the management of cardiac surgery patients. Thirty-two percent of these patients experience preoperative anxiety and 19% experience depression. Patients under the age of 65, as well as those with a longer preoperative stay, are at higher risk [7].

Under normal conditions, the sympathetic nervous system is activated, and cortisol is released in both anxiety and depression. This causes inflammation, which is harmful to the surgical procedure and the patient’s subsequent recovery [8].

Preoperative anxiety causes increased postoperative pain, a greater need for pain medication, and prolonged recovery time [9]. Depression also has an adverse effect on these patients. A meta-analysis found that patients undergoing cardiac surgery who experience perioperative depression are at increased risk of death after surgery [10].

Depression has multiple negative effects on the body and brain, ranging from reduced compliance with treatment to poor self-care [16]. At this point, it should be emphasized that there are specific factors that can influence the likelihood of preoperative psychological problems.

Age is a risk factor for preoperative anxiety. Patients under the age of 65 were three times more likely to experience preoperative anxiety [18]. The most significant risk factor for depression was the preoperative length of hospital stay, especially if it lasted more than 3 days [19].

It is clear that measures should be taken to prevent these psychological problems before surgery, through psychological testing and, where appropriate, timely psychotherapeutic intervention [8].

#### 3.1.2. Postoperative Psychological Complications: Depression—Delirium—PTSD-like Symptoms

Postoperative depression is also a significant problem, with estimates suggesting that 15–25% of cardiac surgery patients experience this emotional disorder [20]. Symptoms can persist for a long time, even months after surgery.

This is not just a temporary emotional reaction, but an independent risk factor for increased use of healthcare resources, poor functional outcomes, and mortality [21]. At the same time, delirium is a postoperative complication that occurs in 20–30% of cardiac surgery patients and has a poorer prognosis.

Postoperative delirium has an acute onset, a fluctuating course, and clinically manifests as cognitive dysfunction, disturbances of consciousness, and attention deficits [22]. It occurs more frequently in elderly patients with preoperative cognitive dysfunction, longer cardiopulmonary bypass duration, and postoperative complications (electrolyte disturbances, fever).

Delirium clearly entails prolonged stays in intensive care units, increases the cost of medical care, and prolongs cognitive dysfunction [23]. It is common for patients to present with clinical symptoms of post-traumatic stress disorder (PTSD), especially those who undergo complex cardiac surgery or remain isolated in the intensive care unit [14].

Patients may relive bad memories and avoid anything related to surgery, which affects their recovery and rehabilitation [11]. The long-term effects of PTSD-like symptoms in cardiac surgery patients require further analysis and investigation [24].

#### 3.1.3. Cognitive and Neuropsychological Outcomes: Post-Hemorrhage Syndrome—Cognitive Impairment

Neurological complications are always investigated after cardiac surgery, as when they occur they have a catalytic effect on patients’ quality of life and hinder their ability to function independently and care for themselves [25]. The condition known as “post-hemorrhage syndrome”—which until recently was thought to be brain damage caused by cardiopulmonary bypass—includes a bunch of cognitive deficits, like trouble focusing, memory loss, and problems with executive function [26].

Recent research shows that neurocognitive decline after cardiac surgery is a multifactorial issue involving the effects of anesthesia, surgical trauma, possible inflammatory reactions, and pre-existing risk factors [27]. The postoperative persistence of cognitive dysfunction varies, with studies reporting rates ranging from 30–80% at discharge and 10–30% six months after surgery, depending on how and when it is assessed [28].

Risk factors for developing these cognitive difficulties include pre-existing cognitive impairment, advanced age, diabetes mellitus, and prolonged surgery times [29]. Recent research has investigated biomarkers—such as light neurotrophic protein—that are thought to aid in the prediction and monitoring of cognitive changes, prompting timely and effective interventions [30].

Long-term cognitive deficits vary greatly, with some patients recovering rapidly, while others experience permanent deficits. Even rarer is the possibility of cognitive improvement compared to their preoperative condition [25].

Preoperative depression causes long-term cognitive decline, emphasizing that mental health management sometimes has a neuroprotective effect. Research is ongoing into individual risk factors for the development of neuroprotective strategies [31].

#### 3.1.4. Social Determinants of Health: Socioeconomic Status—Family Support—Cultural Background

Social determinants are not adequately recognized in standard clinical assessments, even though they play a critical role in the outcome of cardiac surgery. The socioeconomic status of patients has been shown to be an important predictor of both immediate and long-term surgical outcomes [31,32].

In-hospital mortality is higher in patients from lower socioeconomic strata, and medium-term survival rates are also poorer. This observation is independent of traditional risk factors, indicating that social determinants undoubtedly influence patient outcomes through pathways not captured by traditional clinical variables [32].

Social networks and family support are essential for both long-term adherence to treatment and rapid surgical recovery of patients. Patients who have a stable social support system show clearly improved self-care behaviors, rapid psychological adjustment, and more favorable outcomes in their quality of life. In contrast, inadequate support networks and social isolation are associated with delayed recovery, increased psychological distress, and higher readmission rates [15].

Patients’ cultural background also influences their expectations, adherence to treatment plans, and communication styles, highlighting the need for culturally appropriate care strategies [18]. Factors such as limited health literacy, language barriers, and cultural beliefs about illness and treatment have a catalytic effect on the overall surgical experience and its outcome.

Strategies to address these inequalities are implemented by healthcare systems, including the development of appropriate culturally adapted educational materials, the provision of interpretation services, and the creation of community cooperation programs aimed at improving accessibility for different patient population groups [33]. The main psychosocial challenges faced by cardiac surgery patients and their impact on clinical outcomes are summarized in Table 5. An overview of these psychosocial challenges, their underlying pathways, and their impact on clinical and patient-reported outcomes is illustrated in Figure 2.

These findings suggest that psychological distress is not only a comorbidity but also a dynamic determinant of surgical recovery, influencing immune, neuroendocrine, and behavioral pathways that collectively affect outcomes.

### 3.2. Patient-Centered Care Principles

#### 3.2.1. Definition—Theoretical Framework

Patient-centered care is an approach to healthcare delivery that prioritizes the needs, preferences, and values of the individual patient, ensuring that these largely determine all clinical decisions. The Institute of Medicine defines patient-centered care as “the provision of care that respects and responds to the preferences, needs, and values of each patient and ensures that their values guide all clinical decisions.” This definition shifts the focus from a purely biomedical practice to a holistic approach that assesses the full spectrum of human experience associated with disease and its treatment [34]. The studies reviewed demonstrated substantial heterogeneity regarding patient type (elective vs. urgent procedures; CABG vs. valve surgery), intervention format (individual vs. group, in-person vs. digital), and follow-up duration (from one week to twelve months). Accordingly, the following subsections present results by intervention type and indicate the relative strength of evidence (randomized vs. observational).

The World Health Organization adds to this definition, saying that it “consciously embraces the perspectives of families, individuals, and communities and considers them as partners and beneficiaries of advanced health systems.” This definition recognizes that effective healthcare does not simply focus on treating symptoms but addresses the cultural, social, and psychological contexts that shape health [35]. When applied to cardiac surgery, the principles of person-centered care emphasize a holistic approach that recognizes the decisive role of cardiac surgery in prolonging life, shaping relationships, and influencing patients’ future plans. It recognizes that a successful cardiac surgery is not enough if the patient does not regain their ability to care for themselves [36].

#### 3.2.2. Communication—Shared Decision Making

In cardiac surgery care, patient-centered communication is vital, as it includes emotional support, providing information, and making decisions with empathy and compassion. After all, it is an indisputable fact that effective communication reduces anxiety, increases patient satisfaction, and encourages patients to adhere to the treatment plans proposed to them by the medical and nursing staff [37].

In cardiac surgery, patients face unique communication challenges due to the emotional nature of the experience, the complexity of the procedures, and the need to balance information about the risks and potential benefits of heart surgery [38]. Joint decision-making between physician and patient regarding the healthcare to be provided to the patient (based on their data, values, and preferences) is crucial for maintaining a balanced psycho-emotional state before, during, and after surgery [39]. In this case, it may include explaining the available treatment options and predicting their possible outcomes for each patient individually, weighing the risks and benefits of each option. At this point, it should be emphasized that patients need both sufficient time and support to make informed decisions about their care.

Due to the complexity of cardiac surgery, the shared decision-making process may involve multiple steps, decision-makers (possibly including family members), and meetings [37]. It is clear that shared decision-making in cardiac surgery requires organizational support and communication skills.

Clinicians should be trained in effective communication strategies, such as expressing empathy, active listening, and explaining complex medical information in simple language. Appropriate educational handbooks can also be used at this stage to support shared decision-making by providing information that patients can review and discuss with their families [37].

#### 3.2.3. Values—Preferences—Cultural Context of the Patient

Understanding an individual’s beliefs and attitudes toward illness, health, treatment, and quality of life is necessary to accurately determine patients’ values in cardiac surgery care [18]. Patients may prioritize different areas, such as their functional status versus long-term survival, or have specific concerns about disability, impact on family life, and/or pain [14].

These values are greatly influenced by cultural background and can also alter decision-making processes, communication preferences, and expectations regarding family involvement [40]. Cultural factors include religious beliefs, language adaptation, and views on medical interventions [33].

In some cultures, decision-making is family-centered, and individual autonomy is not encouraged, which requires a careful communication approach. Religious beliefs clearly influence views on organ donation, life-sustaining treatment, and the scheduling of procedures around religious ceremonies [41].

Organized healthcare systems are called upon to develop a systematic approach to the cultural identity of patients. In addition, there is an urgent need to train staff to provide interpretation services on cultural competence issues, access to culturally appropriate educational materials, and the formation of balanced partnerships with community organizations that serve diverse population groups [33].

The ultimate goal is to create an environment where patients feel supported, protected, and respected regardless of their cultural background.

#### 3.2.4. Involvement of Family and Caregivers

Family involvement is a key aspect of patient-centered cardiac care, as cardiac surgery is considered to have an impact on the entire family system and not just on the individual patient [42]. Family members may also serve as primary caregivers for the patient during the recovery period, so understanding the treatment plan and self-care requirements is vital [43]. Studies have shown that family-centered care can lead to lower complication rates, shorter hospital stays, and higher patient satisfaction [15].

Family involvement can have many positive effects on cardiac surgery patients, including not only practical assistance with self-care, but also emotional support, advocacy, and participation in medical decision-making [41]. Family members can provide important information about the patient’s preferences, basic functioning, and psychosocial factors that may influence treatment decisions.

In addition, they can serve as important observers of the patient’s condition and recovery, identifying potential problems or complications that might otherwise be overlooked [2]. Family care requires organizational policies and practices that facilitate and support family involvement, such as flexible visiting policies, opportunities for family meetings, caregiver training, and clear communication about family roles and expectations [44].

Healthcare providers should be trained in family assessment and intervention and understand that family relationships can be both a source of support and a source of stress for cardiac surgery patients [44]. The key domains and benefits of a patient-centered approach in cardiac surgery are illustrated in Figure 3.

The convergence of emotional, cognitive, and social factors highlights that patient-centeredness functions not merely as an ethical framework but as a clinical determinant that modulates adherence, resilience, and satisfaction.

### 3.3. Psychosocial Interventions—Patient Support

#### 3.3.1. Preoperative Counseling—Psychological Preparation

A typical example of an evidence-based intervention is preoperative psychological preparation, which applies to a structured systematic approach to patient education in order to manage their expectations and reduce their anxiety. The ultimate goal is the success of the upcoming cardiac surgery.

Empirically documented preoperative interventions are usually multifaceted, combining cognitive restructuring with information provision and behavioral techniques. They work to reduce patients’ psychological barriers, increase their understanding of the upcoming procedure, with the aim of developing adaptive behaviors to manage the psychological stress that overwhelms patients before surgery [45].

The Emotional and Cognitive Education Program (PACE) is an excellent model for preoperative preparation. Among other things, it provides video testimonials from patients, medical instructions from cardiothoracic surgeons, and counseling guidance from specialized mental health professionals. Patients are given information on how to manage their emotional difficulties (fear, anxiety, depression) and cognitive deficits related to surgery. This approach has been shown to result in lower levels of psychological distress and higher positive prognosis scores compared to a control group receiving standard guidance [5]. Hypnosis combined with virtual reality has also been used successfully to reduce preoperative anxiety. Recent research highlights that any virtual reality intervention prior to cardiac surgery significantly reduces both postoperative pain and preoperative stress in patients [46].

Another study highlights that virtual reality combined with hypnosis has been effective in managing fatigue, pain, and anxiety in patients in the cardiac surgery room. This is a low-cost intervention that can work effectively as a supplement to individual counseling therapy [47].

At the same time, there is another approach that includes expectation-focused interventions, which attempt to balance patients’ expectations regarding the entire surgical process and subsequent recovery. Using the randomized controlled PSY-HEART trial, researchers found that psychological interventions prior to cardiac surgery are directly associated with changes in stress hormone responses. Participants who received intervention had reduced increases in cortisol and catecholamines after surgery [12]. This indicates that adequate preparation of patients—on a psychological level—contributes favorably to the regulation of their physiological responses to negative emotions [48].

#### 3.3.2. Role of Members of the Interdisciplinary Team

The interdisciplinary approach to psychosocial issues surrounding cardiac surgery draws on the expertise and skills of social work, psychology, nursing, and other scientific disciplines. Undoubtedly, nurses are at the forefront of psychosocial assessment, as they are the point of contact for patients throughout the surgical procedure. After all, they are responsible for maintaining constant communication with the patients’ families, while also educating them, monitoring their psycho-emotional state, and providing ongoing support [49]. During hospitalization, sleep is a key factor in recovery after critical heart surgery.

At this point, it should be emphasized that nurses can help promote rest and recovery through interventions such as creating a favorable sleep environment and providing supportive care [50]. Psychologists and mental health professionals can offer specialized expertise in psychological assessment, intervention planning, and treatment provision.

The latter are responsible for assessing the psycho-emotional state of patients preoperatively, managing possible anxiety and/or depression in patients with cognitive behavioral therapy, and timely interventions to avoid acute psychological complications [51]. In any case of a multidisciplinary approach to cardiac surgery issues, it is necessary to clarify communication protocols and the roles of each scientist in conjunction with the joint planning of patient care [52].

The social determinants of health can be discussed with social workers, who can indirectly contribute to a good postoperative outcome. These include family dynamics, financial resources, housing stability, and community support networks. Social workers organize hospital discharge, coordinate resources, and manage all practical obstacles that arise and relate to the long-term care and recovery of patients. Their interventions also include supporting caregivers, assisting with benefits, and connecting patients with community resources to ensure ongoing support [53].

Physical therapists aim to promote the psychosocial well-being of patients by enrolling them in early mobilization programs, informing them about the progress and limitations resulting from their activities, and assessing their functional capacity. These interventions boost patients’ self-confidence, helping them regain trust in their physical abilities and develop realistic expectations regarding the time it will take to return to their previous functional status. Finally, the role of occupational therapists is to smoothly regulate patients’ daily activities, assess their cognitive abilities, and develop strategies for adapting to functional limitations [54].

#### 3.3.3. Cardiac Rehabilitation with Psychosocial Components

Psychosocial interventions beyond physical exercise and lifestyle changes include recommendations for restoring normal heart function. These components are considered particularly important parameters for adherence to cardiac rehabilitation programs [55].

Holistic cardiac rehabilitation programs, which take into account both psychological and physical components, show optimal results compared to interventions that focus exclusively on exercise. However, there are also studies that have found that although there was an improvement in physical performance, mental health did not keep pace with physical health without the mediation of targeted psychosocial interventions [13,56].

Another avenue for mutual support among patients with similar conditions is group psychosocial interventions. The latter offer stress relief to participants, facilitating their adaptation in order to achieve optimal learning experiences. Of course, personalized counseling programs [57] can also be implemented in conjunction with specialized spiritual and psychological support programs [58].

Recent studies show improvement in quality of life, mental health, and patient commitment to treatment after exposure to appropriate psychosocial interventions. A typical example is the finding of significant improvements in the prognosis and recovery of patients’ psychological distress in a systematic review of psychological interventions in cardiac surgery patients [59].

It is clear that further research is needed to accurately determine the optimal methodology and duration of psychosocial interventions in cardiac rehabilitation [60]. Technology is being integrated at an ever-increasing rate and is enhancing psychosocial support for patients with new methods. Telemedicine is a field that provides remote access to counseling and monitoring of patients’ mental health. To this end, mobile phone applications have been established to monitor patients’ moods, provide educational resources, and offer stress management tools throughout the course of their recovery [60].

#### 3.3.4. Digital Health—Telemedicine Support

Remote psychological counseling sessions are facilitated by telemedicine platforms, while geographical restrictions are removed and patients’ access to mental health support services becomes more immediate. Telemedicine makes group, family, and individual sessions easier and more immediate for the average cardiac surgery patient who may have reservations about communicating with a psychotherapist in person [60].

At the same time, self-management tools have been developed and promoted that monitor mood, remind patients to take their medication, suggest anxiety reduction techniques, and ensure access to educational resources. In addition, telemedicine offers 24 h support every day, which can save patients in urgent situations of psycho-emotional distress. The ability to integrate telemedicine applications into mobile devices (e.g., cell phones, tablets, laptops) is extremely important, as it ensures continuous monitoring and vigilance of the mental health of cardiac surgery patients [61]. Artificial intelligence (AI) applications are an emerging area of psychological care, offering the possibility of optimizing and personalizing interventions.

AI-based systems process data from patient reports and behavior to predict potential psychological complications and suggest solutions for each patient. In particular, machine learning algorithms predict depression in cardiovascular disease with high accuracy. Artificial-intelligence models have demonstrated moderate predictive performance for postoperative depression, with reported accuracies around 62% and area-under-the-curve (AUC) values between 0.68 and 0.72 [62]. These retrospective studies remain preliminary and highlight challenges related to limited sample sizes, privacy, and interpretability, underscoring the need for prospective validation.

VR applications are a tool for psychological support for patients not only preoperatively but throughout their hospitalization and recovery [17]. Immersive VR environments undoubtedly contribute to teaching relaxation techniques, diverting attention from painful stimuli, and treating anxiety through exposure [46]. As this method is under development, it may become more accessible and can be integrated into the daily clinical care of cardiac surgery patients [63]. Table 6 provides an overview of evidence-based psychosocial interventions in cardiac surgery, highlighting their scope, reported benefits, and supporting evidence.

Although the reported psychosocial interventions consistently demonstrate beneficial effects on anxiety, pain, and recovery trajectories, most studies remain limited by methodological constraints. The majority are small-scale or single-center trials with heterogeneous designs, variable follow-up durations, and nonstandardized outcome measures. Only a minority employed blinding or long-term endpoints, which may affect internal validity and generalizability. Consequently, the evidence base, while encouraging, should be interpreted as preliminary. Larger multicenter randomized controlled trials with standardized psychosocial endpoints are warranted to confirm and expand these findings.

The diversity of interventions—ranging from psychotherapy and mindfulness to digital support—reflects an expanding but fragmented field, underscoring the need for standardized, multidisciplinary protocols.

### 3.4. Impact on Clinical Outcomes and Outcomes Reported by Patients

#### 3.4.1. Short-Term Outcomes: Length of Stay—Speed of Recovery—Complications

The effectiveness of psychosocial interventions is assessed by evaluating specific postoperative conditions, including length of hospital stay and recovery, quality of recovery, and postoperative complications. It has been shown that each intervention results in a significant improvement in prognosis scores 6 weeks after discharge from the hospital, compared to patients receiving standard care. These scores are derived from composite measures that assess recovery at the psychological, social, and physical levels. Consequently, the observed improvements are the result of the simultaneous effect of appropriate interventions on numerous recovery pathways [30].

Compared to the control group, patients receiving psychosocial support preoperatively have better pain scores (primary outcome) and reduced use of analgesic drugs postoperatively (secondary outcome). A recent meta-analysis, which synthesized the results of multiple clinical trials, concluded that psychological interventions are directly associated with up to 33% lower reported pain [64]. Moreover, psychological and social interventions have been shown to reduce self-reported pain by up to 33%, according to pooled results from randomized controlled trials included in the Cochrane review by Ziehm et al. (2017) [11], which analyzed visual-analog-scale (VAS) outcomes across seven studies.

At the same time, it prevents the onset of delirium, a potentially dangerous acute disorder of an individual’s cognitive functions. Its prevention without medication in hospitalized cardiac surgery patients is ensured by timely mobilization, the implementation of healthy sleep programs and reorientation protocols, as well as the involvement of the family in patient care. Optimizing the environmental conditions to which patients are exposed, such as maintaining the day-night cycle and reducing noise can enhance patients’ cognitive functions [65]. It is clear that patients who receive psychological preparation and support both preoperatively and postoperatively have shorter hospital stays, recover more quickly, and experience fewer complications.

As a result, patient satisfaction improves, healthcare costs are reduced, and a more efficient use of resources is ensured [36]. Postoperative symptoms such as depression and anxiety increase the risk of postoperative complications (e.g., atrial fibrillation) following cardiac surgery [66].

#### 3.4.2. Long-Term Results: Quality of Life—Adherence to Treatment—Return to Daily Life

As mentioned above, in general, the functional and psychological status of patients, as well as their quality of life, are clearly better compared to patients who receive standard medical care [67]. At the same time, the prognostic factors of psychological distress in the patient’s recovery environment (usually at home) after cardiac surgery must be taken into account [68].

Recognized scales that assess patients’ quality of life (e.g., the SF-36) show that those receiving psychological support score better on measures of mental and physical health. Patients who were living in poor conditions from the outset benefit most from these interventions [56].

Functional and psychological recovery after cardiac surgery is particularly important, especially for older patients [69]. Similarly, adherence to treatment recommendations (e.g., medication), guided lifestyle changes, and regular reassessment promote recovery and postoperative well-being in cardiac surgery patients.

At this point, it should be emphasized that both perioperative and preoperative predictors of depression are crucial for avoiding late psychological complications that may be long-lasting [70]. The return of patients to their daily lives is another important parameter. Patients who receive holistic care return more quickly to all areas of their social life.

Furthermore, they report greater satisfaction with their family and friends, with their lives in general, while their productivity and self-esteem are rated highly. After surgery, these patients seemed willing to engage more actively in activities that had previously caused them anxiety [54].

Finally, it should be noted that relatives’ beliefs about surgical risk play a decisive role in cardiac surgery, as they influence the adjustment of patients and their caregivers in view of the impending ordeal, but also postoperatively [71].

#### 3.4.3. Patient-Reported Outcome Measures (PROM)

Patient-reported outcome measures (PROM) are becoming increasingly popular for assessing the impact of psychological and social interventions on patients’ health outcomes following cardiac surgery [72]. These self-reporting tools assess aspects of quality of life, functional status, and symptoms that may not be captured by traditional clinical outcome measures.

Some of the most widely used PROMs applied to cardiac surgery patients are the Rose Dyspnea Scale, the Patient Health Questionnaire, and the Seattle Coronary Questionnaire [72]. It should be noted that patient-reported outcomes tend to be more adverse during the first 30 days after surgery, but show significant improvement over time.

It is particularly interesting that the majority of patients show better results 1–2 years after surgery compared to their pre-surgery condition [67]. Elderly patients over 75 years of age show greater improvement compared to younger patients, both in terms of SF-12 physical and emotional component scores and in the frequency of postoperative angina [67]. The use of technology platforms for collecting PROMs and integrating them into digital health records greatly facilitates the intensive monitoring of patient health outcomes [73]. A conceptual framework linking psychosocial challenges, patient-centered interventions, and improved outcomes in cardiac surgery is illustrated in Figure 4.

While the patient-centered model has been widely endorsed, its translation into daily cardiac surgical practice often encounters practical barriers. The urgency and high procedural standardization that characterize cardiac surgery may limit the time available for individualized communication, shared decision-making, and psychosocial support. Institutional constraints, workflow pressures, and hierarchical team structures can further hinder consistent application of patient-centered principles. For example, studies have reported that despite surgeons’ positive attitudes toward holistic care, preoperative discussions frequently focus on technical aspects, with limited attention to emotional readiness or patient values [57,64,70]. These discrepancies highlight that achieving true patient-centeredness requires not only attitudinal change but also organizational adaptation, staff training, and supportive institutional policies.

Interactions between psychosocial status, comorbid conditions, and healthcare access illustrate that recovery trajectories are shaped by an interplay of biological and social determinants, not isolated variables.

### 3.5. Challenges and Barriers

#### 3.5.1. Limited Resources—Staff

Resource constraints often relate to the staff who work in mental health services. At the same time, smaller or resource-constrained institutions lack sufficient expertise in social and psychological interventions in cardiac surgery [74]. Staff shortages (in terms of numbers and training) disrupt the initial screening of patients and hinder the implementation of psychosocial interventions and the ongoing monitoring of patients’ health.

Although nurses working in cardiac surgery clinics are adequately trained to provide psychosocial care to patients, the ability to conduct in-depth psychological assessments is often limited by other more important assessment responsibilities. The rapid turnover of patients and high levels of acuity in cardiac surgery units make it difficult to implement appropriate psychosocial interventions. The fact that there are restrictions on the reimbursement of psychological support services in medical settings is an obstacle to their sustainability. Moreover, these services are the first to face cuts when resources are scarce [74]. Ongoing supervision and training require resources and time that may be hampered by increased clinical demands.

#### 3.5.2. Lack of Standardized Psychosocial Assessment Tools

In cardiac surgery, there are various psychological assessment tools but no standardized screening tools. As a result, practices vary between providers and institutions, leading to the underdiagnosis of patients who would benefit from appropriate psychosocial interventions [74,75]. An innovative tool that aims to provide a comprehensive series of psychological assessments of patients—but which has not yet been widely adopted—is the psychocardiogram.

Adapting screening tools to populations with different cultural backgrounds is a new challenge, as descriptions of psychological symptoms may vary across different cultural groups. There may be a lack of training or technical infrastructure to integrate screening tools into clinical workflows and electronic health records. It is clear that the lack of systematic procedures for implementing screening tools makes patient follow-up inconsistent. It is also obvious that there must be clear protocols for evaluating screening results, but this is often not feasible or consistently implemented in clinical practice [75].

#### 3.5.3. Variation Between Health Systems

Marked differences in practices between different health systems can lead to inequalities in patient care. Large academic medical centers have experienced multidisciplinary teams, in contrast to smaller community health centers, which often lack psychological services [32].

Geographical variations within the same healthcare system may further increase existing inequalities. A typical example is underserved rural communities that do not have direct access to mental health facilities or do not have the resources to serve them, thus limiting these patients’ access to psychological services before, during, and after surgery [60].

Telemedicine appears to fill this gap, although it requires logistical infrastructure that is not accessible in all areas [60]. Healthcare systems with a high degree of commitment to quality indicators and personalized care for each patient invest in psychosocial services. Conversely, systems that emphasize patient volume and efficiency may not consider psychosocial interventions to be of sufficient importance [74].

Quality assessments and certification requirements may vary between different geographical areas and healthcare systems. This may weaken external pressure to implement these programs [74].

#### 3.5.4. Resistance to Interdisciplinary Approaches

Cultural differences arise within healthcare centers that can act as a barrier to the implementation of interdisciplinary approaches. Traditionally, medicine emphasizes specialized knowledge and professional autonomy.

Interdisciplinary collaboration practices are more modern and have recently become widespread. Doctors are sometimes reluctant to take psychologists’ advice into account and often consider psychosocial interventions to be outside their field of expertise [73,74,75].

Among the various medical and paramedical professions, there are communication patterns and power dynamics that undoubtedly shape the acceptance and integration of psychosocial interventions into treatment planning. Historically, social workers and psychologists have been marginalized by medical staff, which affects the power dynamics that are being formed. At the same time, the pressure for productivity and time constraints that characterize healthcare environments also hinder interdisciplinary collaboration. When the focus—especially among medical staff—is on maximizing clinical performance, it is obvious that meetings of members of an interdisciplinary team may seem burdensome or utopian [74].

In any case, without explicit agreements regarding the responsibilities and authority of each specialty or even each member of the interdisciplinary team, there will be chaos in communication and uncertainty regarding the division of responsibilities for general supervision and patient care. This can lead to overlapping responsibilities, gaps in care, or even professional disputes that hinder the smooth collaboration of healthcare professionals [74,75].

The persistence of these barriers reflects broader systemic and cultural challenges within healthcare organizations. Structural causes include resource limitations, fragmented communication across disciplines, and institutional priorities that emphasize procedural efficiency over psychosocial integration. Cultural norms within surgical teams may also reinforce hierarchical decision-making and undervalue emotional aspects of care. To address these issues, targeted strategies such as incorporating psychosocial indicators into quality metrics, implementing interprofessional training in communication and empathy, and fostering organizational cultures that reward holistic care have been proposed. Moreover, embedding psychosocial specialists within surgical teams and integrating digital follow-up tools may help operationalize patient-centered approaches even in resource-constrained settings.

## 4. Future Directions

### 4.1. Integration of Psychosocial Indicators into Surgical Pathways

This process involves the creation of protocols and the development of standardized assessment tools. The ultimate goal is to use them in a variety of healthcare settings and extract clinically useful information for enhanced recovery after surgery (ERAS) protocols [76].

Quality indicators and performance assessments include measures of psychological well-being, traditional clinical outcomes, psychosocial functioning assessments, and patient-reported outcomes. Healthcare systems need to be revamped, as surgical success is not only considered to be the absence of complications and survival, but also the subsequent improvement in patients’ physical and mental well-being [76].

Electronic psychosocial functioning assessments incorporate psychosocial screening tools and represent the future of social and psychological support following cardiac surgery. Standardized assessment templates, automated reminders, and holistic approach planning tools promote the simplified delivery of psychosocial health services [75,76].

Automatic identification of patients at high risk for psychological complications is achieved by clinical decision support systems [67]. Sustainable compensation is achieved through the reduction of postoperative complications, the high cost-effectiveness of psychosocial interventions, the promotion of quality of life, short hospital stays, and improved long-term outcomes of cardiac surgery [74].

### 4.2. Personalized Medicine—Mental Health

Undoubtedly, the future of psychosocial care in cardiac surgery will be determined by developments in personalized medicine. Genetic and genomic factors are areas of active research and influence mental resilience, stress response, and ultimately treatment response [75,76].

Pharmacogenomic testing influences the dosage and type of antidepressants administered to cardiac surgery patients. An emerging area of research is the identification of biomarkers that are directly related to favorable psychological outcomes for patients [76,77].

The identification of patients at high risk of psychological complications is facilitated by blood biomarkers that assess inflammation, stress response, and neuroplasticity. A typical example is the sRAGE biomarker, which is found in cardiac surgery patients with major depression [77].

In this case, machine learning algorithms contribute to the analysis of complex patterns of clinical, demographic, social, and psychological factors that predict individual risk profiles, while also promoting individually tailored intervention strategies. These predictive models help to normalize the distribution of resources by prioritizing high-risk patients. Personalized interventions based on the characteristics, response patterns, and preferences of each patient are another area under development [75,76,77].

Some patients respond well to cognitive-behavioral approaches. For others, mindfulness-based interventions and peer support groups work better. In any case, clarifying these differences, in combination with the creation of algorithms that will make decisions on the appropriate interventions, will have a catalytic effect on enhancing the efficiency and effectiveness of psychosocial care programs [52].

### 4.3. The Role of Artificial Intelligence and Digital Platforms in Psychosocial Care

The rapid expansion of artificial intelligence (AI) applications is shaping broad prospects for improving the social and psychological care of cardiac surgery patients. AI-based screening tools can analyze data from electronic health records (EHR), patient-reported outcomes (PRO), and behavioral receptors, identifying those at high risk of psychological complications. At the same time, natural language can be processed by algorithms that are applied during communication between doctors and patients and can detect even small changes in cognitive functions, mood, and/or anxiety experienced by heart disease patients who are candidates for surgery [78]. Machine learning models apply predictive analysis and predict the course of each patient, suggesting preventive interventions that are targeted to each patient [62,77,78].

These systems have the ability to integrate and analyze various data streams, as well as behavioral indicators, PROs, and physiological monitoring data. Continuous, real-time monitoring of patients’ mental health is achieved through smartphone apps and mobile devices, enabling dynamic interventions when worrying deviations are detected [62].

Chatbots and AI virtual assistants offer the ability to respond to individual concerns and provide personalized counseling support, along with management strategies where necessary. Although they cannot replace the benefits of human interaction with a psychotherapist, these systems act as a complement, enhancing psychological support services and providing 24 h availability [79].

Digital therapy platforms also provide training in mindfulness, cognitive behavioral therapy (CBT), and anxiety management techniques in an interactive format, at any time of the day. Its integration with EHR systems offers the possibility of supervision by healthcare providers, while maintaining patient autonomy [79]. The integration of digital platforms, AI-driven decision support, and remote care services into a connected ecosystem is depicted in Figure 5.

### 4.4. Policies to Promote Holistic Cardiac Surgery with a Focus on the Patient

Contemporary healthcare policies must support holistic models of personalized cardiac surgery approaches to patients. In these models, psychosocial care is a fundamental part of patient care and not merely an optional extra. Of course, this ambitious plan requires clear guidelines and not just vague statements about psychosocial benefits [80]. Certification standards for cardiac surgery programs must meet certain criteria relating to staff training, psychosocial intervention capabilities, and monitoring of patient health [74].

Both procedural measures (provision of interventions, screening rates, monitoring of results) and structural elements (resources, staff, protocols) must be taken into account. Public disclosure of psychosocial care quality indicators and their regular evaluation promote the continuous improvement of the services provided [74,75].

Policies should cover the costs of interdisciplinary team activities and fully cover mental health services in medical settings. Value-based payment systems should incorporate patient-reported outcomes and psychosocial indicators as quality indicators that influence reimbursement rates [74].

To achieve this goal, medical and nursing schools are called upon to incorporate the principles of personalized healthcare and psychosocial intervention into their core curriculum while ensuring that health professionals are continuously updated on their skills in the above-mentioned areas [81].

## 5. Discussion

This study makes it clear that psychosocial care focused on cardiac surgery patients is based on the assessment of the complex interaction between medical, social, and psychological factors. A holistic approach to cardiac surgery patients yields optimal clinical outcomes in terms of satisfaction with the surgical experience and subsequent quality of life, compared to those receiving standard medical care [52]. Overall, 26 of the studies included in this review were randomized controlled trials or systematic reviews, while the remainder were observational or qualitative in nature. As most RCTs were single-center and modest in size, the strength of causal inference remains limited. Findings should therefore be interpreted as associative rather than definitive, highlighting the need for larger, multicenter confirmatory studies.

The body of evidence reviewed reveals both consolidated and emerging domains. Findings from systematic reviews and multicenter RCTs provide robust support for the beneficial effects of psychosocial interventions on anxiety, depressive symptoms, and perceived quality of life, whereas evidence regarding long-term functional outcomes, cost-effectiveness, and digital applications remains preliminary. This distinction underscores the need to interpret current results within their methodological context while recognizing areas where future research is rapidly evolving.

The lack of standardized screening tools, limited resources, practical and systemic differences, and the resistance often encountered in interdisciplinary collaboration are obstacles to the effective implementation of these interventions. However, these are not insurmountable obstacles but challenges that can be addressed through programs promoting a new culture and through the development of strategic policies to achieve these goals [82].

Despite the encouraging results of psychosocial interventions in cardiac surgery patients, there are still gaps in research regarding cost-effectiveness, standardized intervention protocols, and long-term outcomes. Intensive research and randomized multicenter controlled trials are needed to address the challenges of widespread implementation of these practices in healthcare settings [44].

Psychosocial intervention in cardiac surgery care is not a utopia achievable only by medical institutions of exceptional performance. On the contrary, all institutions offering cardiac surgery care services must seek the necessary resources and demand the appropriate support in order to apply psychosocial intervention techniques to a wide range of patients. Psychosocial interventions have been associated with reduced morbidity and may contribute to lower healthcare costs.

As healthcare systems move towards adopting evidence-based care techniques and personalized approaches to surgical patients, it is clear that providers who devote effort, money, and time to offer a holistic approach to their patients will find themselves in a privileged position of competition in terms of quality [56]. Emerging data indicate that such programs may improve survival, though confirmatory multicenter trials are needed.

Of course, this cannot be achieved overnight, but requires an ongoing commitment from political leaders and those responsible for formulating healthcare optimization policies. At the individual level, every healthcare professional in the field of cardiac surgery must approach the term “success” differently and redefine the goals of cardiac surgery [80].

Despite growing evidence on the importance of psychosocial care in cardiac surgery, several gaps remain. First, there is a lack of standardized screening tools to systematically identify anxiety, depression, and other psychosocial risk factors in surgical patients. Second, most available studies are single-center with small sample sizes, limiting generalizability and making it difficult to establish clear evidence-based guidelines. Third, few randomized controlled trials directly evaluate the effectiveness of psychosocial interventions on clinical outcomes such as complications, readmissions, or long-term survival. In addition, there is limited research on the cost-effectiveness of integrating psychosocial programs into routine surgical care, which is crucial for policy-making and resource allocation. Finally, emerging areas such as digital health interventions, artificial intelligence–based support, and culturally tailored care are underexplored and warrant robust investigation to guide future practice.

Several low-cost, scalable psychosocial approaches have proven feasible and effective in cardiac-surgery populations, including structured telephone counseling [57], brief mindfulness-based sessions [17], and community or lay-support models integrated into rehabilitation [54]. Such strategies facilitate equitable access to psychosocial care even in resource-constrained settings.

Despite growing evidence linking psychosocial care to improved recovery, major knowledge gaps persist. Based on this review, the following five areas represent key research priorities:1.Multicenter randomized controlled trials testing standardized psychosocial care bundles (e.g., combined CBT, social support, and education).2.Cost-effectiveness analyses and health-economic modeling to quantify financial impact and support policy integration.3.Implementation studies in low-resource and community settings, ideally using stepped-wedge or cluster-RCT designs to assess scalability.4.Validation of standardized psychosocial screening tools across diverse populations and healthcare systems.5.Prospective validation of artificial-intelligence–based predictive models, with reporting of sensitivity, specificity, and AUC to ensure clinical reliability.

Addressing these priorities would strengthen the evidence base and facilitate the translation of psychosocial care into routine cardiac surgical practice.

In summary, the ultimate goal of cardiac surgery is not simply to perform the procedure successfully and avoid postoperative complications, but also to treat patients with understanding, assessing the psycho-emotional challenges that arise after such an ordeal, providing them with psychosocial support, and improving their quality of life in the long term.

## 6. Limitations

This review has several limitations that should be acknowledged. First, as a narrative review, it does not follow the strict methodological framework of a systematic review and therefore may be subject to selection bias in the identification and inclusion of studies. Although an extensive search strategy was applied across multiple databases, some relevant publications—particularly unpublished studies, gray literature, or articles in languages other than English—may not have been captured. The literature screening process was performed independently by two reviewers, and disagreements were resolved through discussion and consensus. As this was a narrative review rather than a formal systematic review, no structured risk-of-bias or quality appraisal tool was applied. Consequently, the findings should be interpreted descriptively, with attention to the inherent variability of the included studies. Only English-language studies were included, which may have led to the exclusion of relevant non-English publications and introduced potential language bias.

Second, the heterogeneity of study designs included (ranging from randomized controlled trials to observational and qualitative studies) complicates direct comparison and limits the ability to draw firm conclusions regarding causality. The variability in outcome measures, especially for psychosocial parameters such as anxiety, depression, and quality of life, also makes it difficult to establish standardized benchmarks.

Third, many of the included studies were conducted in high-resource healthcare systems, which may limit the generalizability of findings to settings with fewer resources or different cultural contexts. Psychosocial care and patient-centered interventions are strongly influenced by local health policies, sociocultural norms, and available infrastructures, which may not be fully reflected in the literature reviewed.

Finally, the time frame of the included studies (2009–2025) ensures a contemporary focus, but longer-term historical perspectives and earlier foundational work may have been underrepresented. Similarly, rapid developments in digital health and artificial intelligence–based psychosocial interventions mean that some of the most recent innovations may not yet be supported by robust empirical evidence. Several of the most recent references (2024–2025) include digital health and artificial intelligence studies published as preprints or early-phase protocols. This reflects the rapid evolution of the field, where technological progress often outpaces the accumulation of mature clinical outcome data. Therefore, conclusions regarding digital interventions should be considered preliminary and subject to future validation. A small number of included studies were early-phase trials or preprints without formal peer review; these were retained for completeness but are discussed with appropriate caution, especially when supporting emerging digital health evidence.

These limitations may reduce the generalizability and internal validity of the conclusions, as smaller, heterogeneous studies are more prone to sampling bias and uncontrolled confounding. Future investigations employing multicenter, adequately powered designs, standardized psychosocial measures, and rigorous bias control could strengthen the evidence base and mitigate these constraints. Despite these limitations, this review provides a comprehensive synthesis of current knowledge and highlights key directions for future research and clinical practice.

## 7. Conclusions

Psychosocial and patient-centered approaches are essential to achieving optimal outcomes in cardiac surgery. Beyond technical success, addressing anxiety, depression, cognitive impairment, and social support improves recovery, reduces complications, and enhances long-term quality of life and treatment adherence. Integrating patient-centered principles—such as shared decision-making, family involvement, and culturally sensitive care—together with emerging tools like telemedicine and digital platforms, represents a necessary shift toward holistic care. Policymakers should consider supporting reimbursement pilots for psychosocial and digital support interventions, coupled with economic evaluations to determine their cost-effectiveness and real-world value. Success in cardiac surgery should therefore be redefined to require not only patient survival but also the restoration of their well-being, dignity, and overall quality of life. Future implementation efforts should be accompanied by concurrent evaluation using implementation science frameworks and cost-effectiveness analyses to ensure evidence-based integration into healthcare systems.

## 8. Key Messages

Psychosocial factors such as anxiety, depression, cognitive impairment, and social support significantly influence outcomes after cardiac surgery.Patient-centered care, including shared decision-making and family involvement, improves recovery, treatment adherence, and quality of life.Telemedicine and digital health tools expand access to psychosocial support and enhance continuity of care.Success in cardiac surgery should be redefined to require not only patient survival but also the achievement of long-term well-being, dignity, and quality of life.

## Figures and Tables

**Figure 1 healthcare-13-02957-f001:**
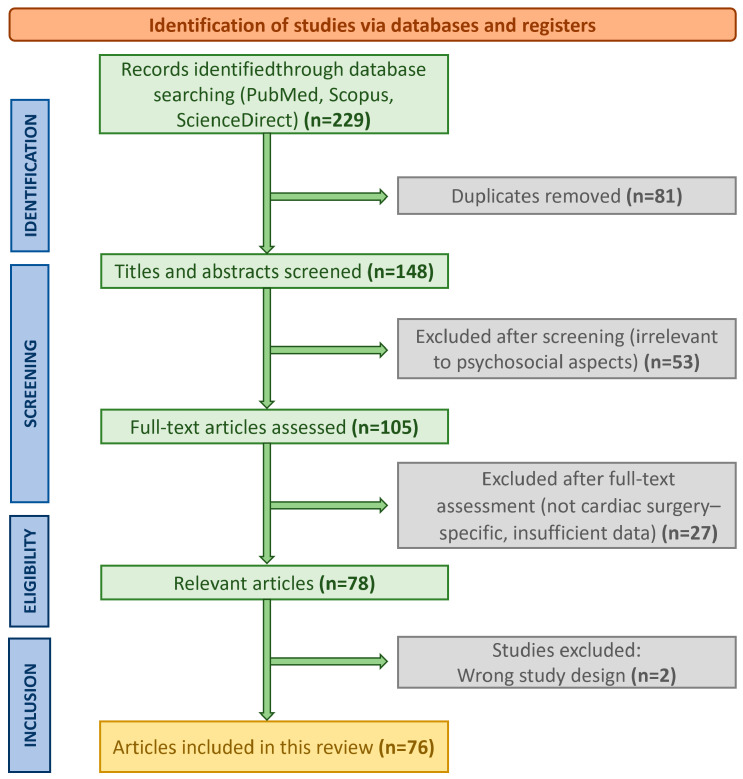
PRISMA-style flow diagram illustrating the selection process for studies included in this narrative review. Although this is not a systematic review, a structured search strategy was applied across PubMed, Scopus, and ScienceDirect to ensure transparency and reproducibility of the literature identification and screening process.

**Figure 2 healthcare-13-02957-f002:**
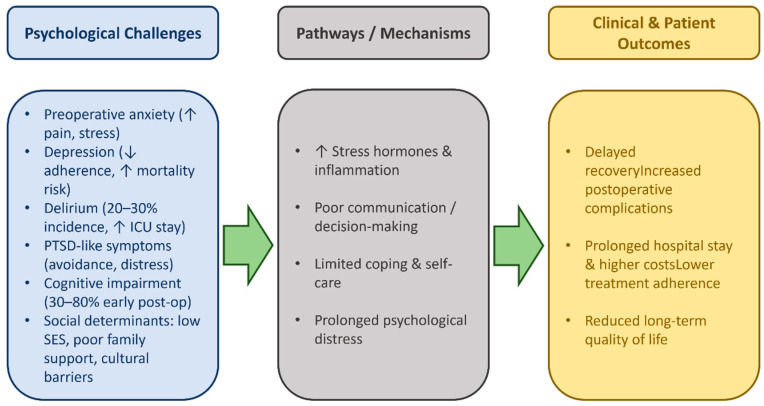
Psychosocial challenges and their impact on outcomes after cardiac surgery. Flow diagram summarizing key psychosocial challenges—preoperative anxiety, depression, delirium, PTSD-like symptoms, cognitive impairment, and adverse social determinants of health—and the pathways through which they affect stress responses, coping, and decision-making. These factors contribute to delayed recovery, higher complication rates, longer hospital stay, lower treatment adherence, and reduced long-term quality of life. Evidence-based associations are supported by findings from studies summarized in Table 2 and Table 3, whereas interconnections shown with dashed arrows represent the authors’ conceptual synthesis.

**Figure 3 healthcare-13-02957-f003:**
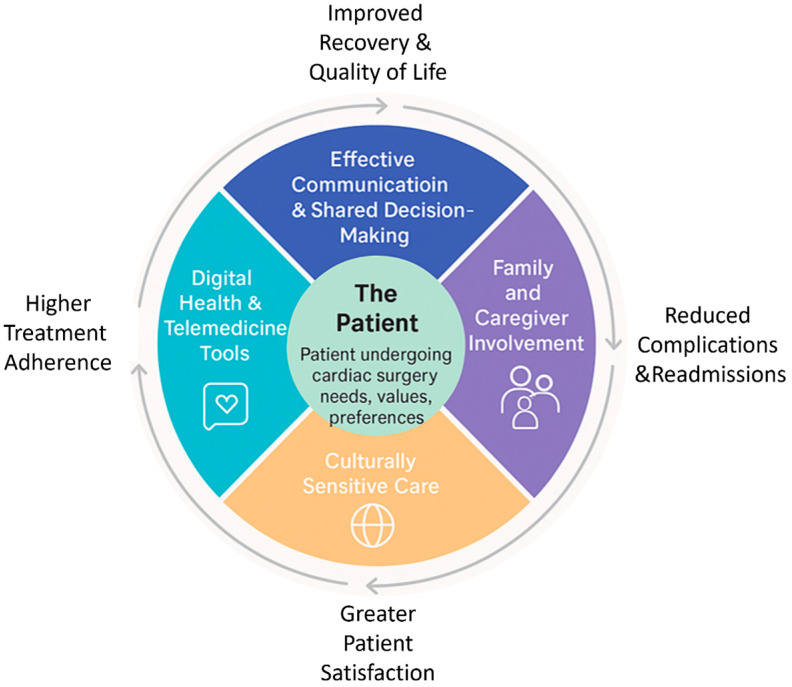
Patient-centered care framework in cardiac surgery. Circular conceptual model depicting the patient at the center—emphasizing individual needs, values, and preferences—surrounded by five core domains of patient-centered care: effective communication and shared decision-making, family and caregiver involvement, culturally sensitive care, multidisciplinary support, and digital health/telemedicine tools. Arrows highlight how these domains collectively contribute to improved recovery and quality of life, reduced complications and readmissions, higher treatment adherence, and greater patient satisfaction. Framework elements in solid boxes are evidence-derived (see Table 3), while shaded areas reflect the authors’ conceptual integration.

**Figure 4 healthcare-13-02957-f004:**
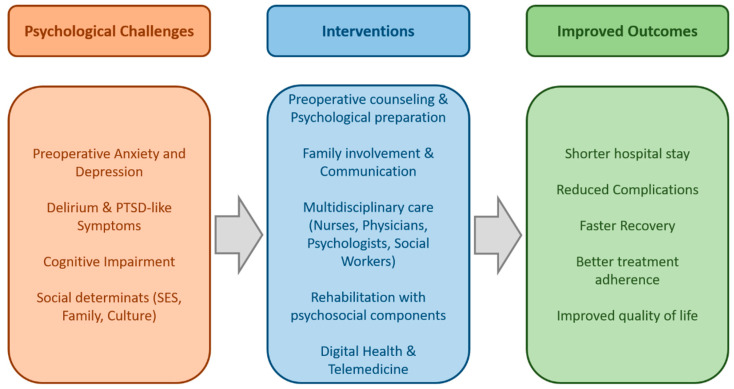
Conceptual framework of psychosocial dimensions in cardiac surgery. Psychosocial challenges such as preoperative anxiety, depression, delirium, cognitive impairment, and social determinants of health adversely affect surgical recovery. Evidence-based interventions—including preoperative counseling, family involvement, multidisciplinary care, rehabilitation programs, and digital health solutions—can mitigate these challenges. Their integration into patient-centered care contributes to improved outcomes, including faster recovery, reduced complications, enhanced treatment adherence, and better long-term quality of life. Interventions are supported by RCTs or systematic reviews (Table 3); others indicate emerging or observational evidence.

**Figure 5 healthcare-13-02957-f005:**
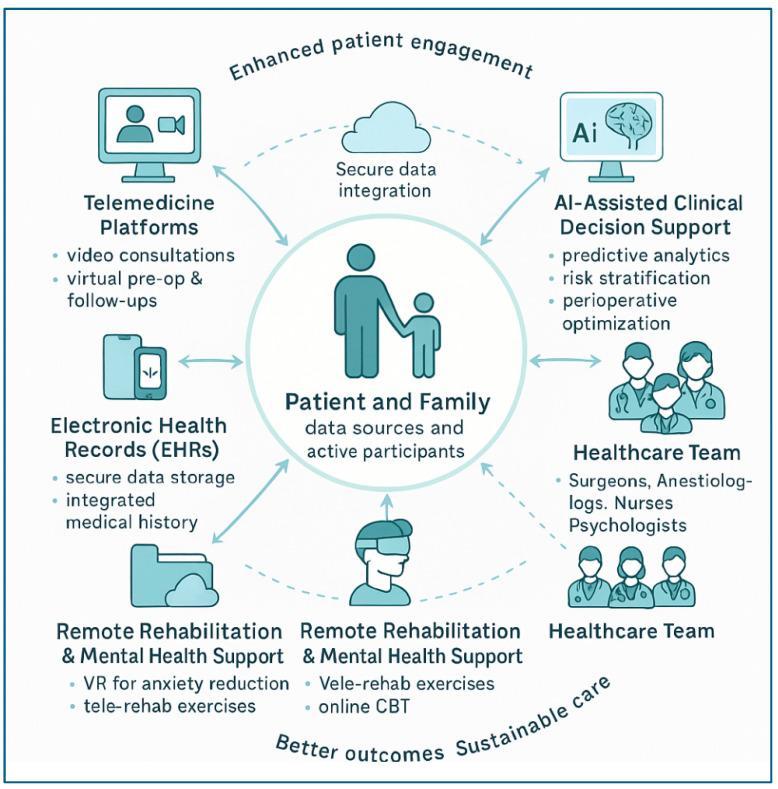
Digital health and AI ecosystem supporting patient-centered cardiac surgery care. Network-style diagram illustrating the central role of the patient and family as active participants in data sharing and decision-making. Surrounding components include telemedicine platforms, mHealth applications and wearables, electronic health records (EHRs), AI-assisted clinical decision support, healthcare team collaboration, and remote rehabilitation/mental health services. Bidirectional arrows indicate continuous data flow and feedback loops, enabling secure data integration, predictive analytics, personalized interventions, and seamless communication to enhance patient engagement, improve outcomes, and support sustainable care. Pathways highlighted correspond to evidence from meta-analyses (e.g., [1,10,11]); dashed lines depict proposed mechanisms requiring further validation.

**Table 1 healthcare-13-02957-t001:** Data extraction fields for included studies. This table summarizes the main data items extracted from each eligible study during the review.

Data Item	Description
First author and year	Name of the first author and year of publication of the study.
Country/setting	Country (and, where available, type of healthcare setting) in which the study was conducted.
Study design	Study type (e.g., randomized controlled trial, prospective cohort, retrospective cohort, cross-sectional study, systematic review, meta-analysis, qualitative study).
Population/surgery type	Description of the study population (e.g., adults after CABG, valve surgery, mixed cardiac surgery population) and type of procedure.
Sample size	Number of participants included in the analysis.
Psychosocial domain(s)	Main psychosocial outcomes assessed (e.g., anxiety, depression, quality of life, social support, cognitive function, PTSD, distress).
Intervention (if applicable)	Type of psychosocial, educational, rehabilitative, or digital intervention evaluated (e.g., psychological preparation, music therapy, VR, e-health, tele-rehabilitation).
Outcome measures	Primary and secondary endpoints, including clinical (e.g., complications, mortality, ICU/hospital length of stay) and psychosocial outcomes (e.g., validated scales for anxiety, depression, QoL).
Effect direction/key findings	Short narrative summary of the main effects (e.g., “intervention group showed reduced anxiety compared to control,” “depression associated with increased mortality”).
Follow-up duration	Time points of assessment (e.g., immediate postoperative period, discharge, 3-month, 12-month follow-up).

**Table 2 healthcare-13-02957-t002:** Summary of methodological quality of included studies. Overall study quality was judged qualitatively using the principal domains of the Cochrane RoB 2 (for RCTs) and Newcastle–Ottawa Scale (NOS) (for observational studies). The majority of RCTs demonstrated low to moderate risk of bias, while observational studies generally showed moderate quality due to limited blinding and sample size.

Reference No.	Author (Year)	Study Type	Quality Tool	Overall Risk/Quality
Protogerou et al. (2015) [1]	Systematic review/meta-analysis	—	High quality	Comprehensive RCT synthesis
Savio & Hariharan (2020) [5]	RCT	RoB 2	Low risk	Clear randomization and outcomes
Rousseaux et al. (2020) [9]	RCT	RoB 2	Low risk	Adequate allocation concealment
Takagi et al. (2017) [10]	Meta-analysis	—	High quality	Quantitative synthesis, no bias concerns
Ziehm et al. (2017) [11]	Cochrane review	—	High quality	Standardized protocol adherence
Salzmann et al. (2017) [12]	RCT	RoB 2	Moderate risk	Small sample, unclear blinding
Sibilitz et al. (2016) [13]	RCT	RoB 2	Low risk	Registered trial, minimal bias
Spindler et al. (2023) [3]	Observational	NOS	Moderate quality	Prospective, limited control
Navarro-García et al. (2011) [8]	Observational	NOS	Moderate risk	Single-center, limited follow-up
Poole et al. (2015) [14]	Observational	NOS	Moderate risk	Small sample
Guo et al. (2025) [15]	Controlled intervention	NOS	Moderate quality	Non-randomized

**Table 3 healthcare-13-02957-t003:** Characteristics of representative studies included in the review.

Author (Year)	Country	Study Design	Sample (n)	Population/Setting	Psychosocial Domain or Intervention	Main Findings/Outcomes
Protogerou et al. (2015) [1]	UK	Systematic review & meta-analysis	2134 (18 RCTs)	Cardiac surgery patients	Psychological interventions (CBT, relaxation, counseling)	Reduced depression and anxiety (SMD −0.43, *p* < 0.01)
Spindler et al. (2023) [3]	Denmark	Prospective observational	108	Postponed elective cardiac surgery	Anxiety, depression, social support	Preoperative distress predicted delayed recovery and complications
Savio & Hariharan (2020) [5]	India	Randomized controlled trial	120	CABG patients	Structured psychosocial support program	Intervention improved QoL and reduced postoperative complications
Rousseaux et al. (2020) [9]	Belgium	Randomized controlled trial	60	CABG and valve surgery	Hypnosis and virtual reality	Significantly reduced anxiety and fatigue; 20% lower pain intensity
Takagi et al. (2017) [10]	Japan	Systematic review & meta-analysis	9201 (10 studies)	Cardiac surgery	Depression or anxiety	Increased mortality risk (OR 1.52, 95% CI 1.12–2.06)
Navarro-García et al. (2011) [8]	Spain	Observational cohort	212	Cardiac surgery ICU	Preoperative mood disorders	Depression predicted longer ICU stay and higher morbidity
Poole et al. (2015) [14]	UK	Observational	210	Post–cardiac surgery ICU	Illness concern and psychological distress	Strong association between distress and prolonged ICU stay
Ziehm et al. (2017) [11]	Germany	Cochrane review	7 RCTs	Cardiac surgery	Psychological interventions for acute pain	33% reduction in pain intensity vs. control
Sibilitz et al. (2016) [13]	Denmark	RCT	210	Valve surgery	Cardiac rehabilitation	Improved physical but not mental QoL at 12 months
Salzmann et al. (2017) [12]	Germany	RCT	56	CABG patients	Preoperative psychological preparation	Lower postoperative cortisol and catecholamine levels
Hill et al. (2024) [16]	Canada	Pilot observational	48	Hypertrophic cardiomyopathy	Patient involvement in care	Improved psychosocial outcomes and satisfaction
Guo et al. (2025) [15]	China	Controlled intervention study	200	CAD patients in rehab	Cardiac rehabilitation + psychological intervention	Reduced exercise phobia; improved adherence

**Table 4 healthcare-13-02957-t004:** Key randomized controlled trials and systematic reviews assessing psychosocial or behavioral interventions in cardiac surgery.

Author (Year)	Study Type	Intervention	Psychosocial Outcomes	Main Quantitative Findings (Effect Size/95% CI)	Evidence Level
Protogerou et al. (2015) [1]	Systematic review & meta-analysis	Psychological interventions	Depression, anxiety	SMD −0.43 (−0.65 to −0.21), *p* < 0.001	High
Takagi et al. (2017) [10]	Meta-analysis	Depression or anxiety (observational data)	Mortality	OR 1.52 (1.12–2.06), *p* = 0.006	High
Ziehm et al. (2017) [11]	Cochrane review	Psychological interventions for pain	Pain intensity (VAS)	↓ 33% mean reduction vs. control	High
Rousseaux et al. (2020) [9]	RCT	Hypnosis + VR	Anxiety, fatigue, pain	↓ anxiety 24%, ↓ pain 20%, *p* < 0.05	Moderate–high
Sibilitz et al. (2016) [13]	RCT	Cardiac rehabilitation	Mental health, QoL	Physical ↑, mental no change	Moderate
Salzmann et al. (2017) [12]	RCT (PSY-HEART)	Preoperative CBT-based preparation	Stress biomarkers	↓ cortisol and catecholamines, *p* < 0.01	Moderate–high
Savio & Hariharan (2020) [5]	RCT	Structured psychosocial support	Depression, QoL	Improved psychological scores (*p* < 0.01)	Moderate
Guo et al. (2025) [15]	Controlled study	Psychological rehab	Exercise phobia	30% reduction in anxiety; ↑ adherence	Moderate
Hill et al. (2024) [16]	Observational	Patient involvement	QoL, anxiety	Improved patient satisfaction	Moderate
Salzmann et al. (2020) [17]	Review	Psychological preparation	Anxiety, recovery	Consistent reduction in perioperative anxiety	Moderate

**Table 5 healthcare-13-02957-t005:** Psychosocial Challenges and Their Impact on Cardiac Surgery Outcomes.

Psychosocial Factor	Prevalence in Cardiac Surgery Patients	Impact on Outcomes	Key References
Preoperative Anxiety	~32%	Increased pain, delayed recovery	[8,9,18]
Depression	15–25%	Higher mortality, poor self-care, prolonged hospitalization	[10,20,29]
Delirium	20–30%	Cognitive dysfunction, longer ICU stay, increased costs	[22,23]
PTSD-like symptoms	Variable	Psychological distress, impaired rehabilitation	[11,14,24]
Cognitive impairment	Up to 30–80% at discharge	Long-term deficits in memory, executive function	[25,26,27,28]
Social determinants (SES, family support, culture)	Not uniformly measured	Delayed recovery, poor adherence, higher readmission	[15,32,33]

**Table 6 healthcare-13-02957-t006:** Evidence-Based Psychosocial Interventions in Cardiac Surgery.

Intervention	Description	Reported Benefits	References
Preoperative counseling (PACE, expectation-focused, VR/hypnosis)	Structured education, cognitive restructuring, relaxation training	Reduced anxiety, improved coping, better physiological stress response	[12,45,46,47,48]
Family involvement	Flexible visiting, caregiver education	Lower complications, shorter hospital stays, higher satisfaction	[41,42,43,44]
Multidisciplinary care (nurses, psychologists, social workers)	Coordinated psychosocial and medical care	Improved recovery, reduced distress, better follow-up	[49,50,51,52,53,54]
Rehabilitation with psychosocial components	Holistic programs (exercise + mental health)	Improved quality of life, adherence, long-term outcomes	[13,55,56,57,58,59,60]
Telemedicine and digital health	Remote counseling, monitoring apps, AI/VR support	Expanded access, continuity of care, 24/7 support	[17,60,61,62,63]

## Data Availability

No new data were created or analyzed in this study. Data sharing is not applicable to this article.

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
