# Peer review of "Patient-Centered Cardiac Surgery: Psychosocial Challenges, Evidence-Based Interventions, and Future Horizons"

_healthcare, 2025, doi:10.3390/healthcare13222957_

Round 1
Reviewer 1 Report
Comments and Suggestions for Authors
Abstract (lines ~23–45)
- Phrases like “both of which are associated with delayed recovery, increased complications, and higher mortality” are stronger than the evidence you surveyed likely supports. Change to “are associated with” or “linked to” and indicate level of evidence (observational vs. RCT).
Edit: replace “both of which are associated with delayed recovery…” → “both of which have been associated in observational studies with delayed recovery…” - The abstract says “narrative literature review … focusing on recently published studies” without dates, search yields, or selection numbers. Add the search period and a sentence like “(searches run to June 2024; X studies included).” This makes the review immediately more credible.
- “reduce reported pain by up to 33%” — which interventions, what study types, and is this pooled? Either add parentheses: “(pooled RCTs/meta-analysis reported up to 33% reduction in self-reported pain)” or tone it down. Provide the supporting reference number in the abstract.
Introduction (lines ~49–83)
- Some sentences restate similar points (psychosocial factors impact outcomes) with minor rephrasing. Tighten to 3–4 short paragraphs: burden/prevalence → mechanisms (why psychosocial matters) → gap in current practice → objective of review.
- You state “one-third … psychological distress” and specific 32%/19% figures; ensure these come from representative, clearly defined studies (which populations? elective vs urgent? single center vs multi). If figures are heterogeneous, present ranges and note heterogeneity.
Edit suggestion: “Prevalence estimates vary: preoperative anxiety ranges from X–Y% (median ~32% in pooled observational studies), while depression ranges from A–B%.” Cite specifics. - You state “most recent research—from the last fifteen years” but methods say 2009–2024 (15 years) — make these match and state in intro: “we review literature 2009–June 2024 with emphasis on interventions with documented evidence.”
Materials & Methods (2.1–2.6, lines ~84–135)
- The manuscript says “narrative review … with emphasis on a systematic approach.” This hybrid phrasing is confusing. Either convert this to a systematic/scoping review (and adopt PRISMA and risk-of-bias methods) or clearly describe it as a narrative review with transparent search methods. If you keep narrative, explicitly state you did not perform formal quality appraisal and state implications.
Action: If keeping narrative, replace “systematic approach” with “structured search strategy” and add transparency (see points below). If converting to systematic, add PRISMA flow, search strings, screening numbers, and risk-of-bias methods. - State how many reviewers screened titles/abstracts, handled disagreements, and what data items were extracted (sample size, study design, outcomes, effect sizes). Add a short table describing the extraction fields.
Results / Thematic Sections (3–6: Psychosocial challenges, Patient-centered principles, Interventions, Outcomes)
- The review reads as a narrative synthesis without an accompanying study characteristics table (author, year, country, design, N, population, intervention, key outcomes). Readers need this to assess generalizability.
Action: Add a “Study characteristics” table and a separate “Key RCTs / systematic reviews” table that summarizes effect sizes and confidence intervals where available. - Example: “psychological and social interventions reduce reported pain by up to 33%” — add the specific meta-analysis citation and clarify whether this comes from RCTs, sample size, and outcome measure (self-reported pain VAS? HRQoL?). I saw the 33% claim in the manuscript but it’s not clear which reference supports it—please add inline citation and brief methodology of that pooled estimate.
- Many interventions (e.g., VR, hypnosis, group therapy) are lumped together; discuss heterogeneity in patient populations (elective vs urgent CABG, valve surgery), intervention intensity, and follow-up durations. Add subsections: Intervention type → evidence strength (RCTs vs observational) → magnitude and duration of effect.
- The manuscript cites AI predicting depression with “high accuracy (62%)” — 62% is not high for predictive models; be cautious and add sensitivity/specificity or AUC if available. Also discuss privacy/implementation and evidence quality for AI—many studies are preliminary or retrospective.
- Figures (1–4) are conceptual and useful, but add short legends, note which elements are authors’ conceptual models vs. evidence-derived, and cross-link to table/paragraph where components are discussed.
- You mention socioeconomic and access issues but give limited practical solutions. Add examples of low-cost scalable interventions and evidence (e.g., telephone counseling trials, lay-worker models).
Discussion (section 9) & Interpretation
- Statements suggesting causality or broad programmatic cost-savings should be rephrased to reflect the evidence base (many RCTs small/single-center). Replace strong phrasing with qualifiers: “evidence suggests,” “emerging data indicate,” or “limited RCT evidence supports…”
- The discussion lists gaps but would be stronger if reorganized into top 5 research priorities (e.g., multicenter RCTs of standardized psychosocial bundles; cost-effectiveness analyses; implementation trials in low-resource settings; standardized screening tool validation; AI predictive model prospective validation). Give specific trial designs (cluster RCT vs stepped-wedge) and payers’ data needs for policy uptake.
- Reiterate how many included studies were RCTs vs observational, and how their methodological limitations affect confidence.
Limitations (section 10)
- State whether study selection was single-reviewer vs dual-reviewer and if any quality appraisal was performed — this is crucial.
- You limited to English; state how that may have excluded key non-English studies.
- Many citations to 2024–2025 preprints/protocols exist; explicitly state that digital health evidence is rapidly evolving and may outpace rigorous clinical outcome data.
Conclusions and Key messages
- Tweak wording to avoid policy prescriptions that aren’t fully grounded in cost-effectiveness data (e.g., “Policies should cover the costs …” → “Policymakers should consider supporting reimbursement pilots coupled with economic evaluation to determine value.”)
- Add a final line that prioritizes evidence generation: “Implement with concurrent evaluation (implementation science / cost-effectiveness).”
References, citations & currency
- Your reference list includes 2025 publications and even items from 2025 in the bibliography—this is acceptable if the journal allows late citations, but ensure the manuscript’s “search through” date (methods: June 2024) matches included references. If you included 2025 work, explicitly state searches were updated on X date. Otherwise remove/justify post-cutoff items.
- A few references (e.g., examples of pain reduction, AI model performance) appear to be preliminary or from non-peer-reviewed sources—flag these and prefer peer-reviewed meta-analyses or RCTs for definitive claims.
- When you assert numerical effects (length of stay, percentage pain reduction), place the exact citation(s) in the sentence and, if possible, include the study design (RCT vs observational).
Author Response
Dear Reviewer 1,
Reviewer 1 Comment 1: Abstract (lines ~23–45)
- Phrases like “both of which are associated with delayed recovery, increased complications, and higher mortality” are stronger than the evidence you surveyed likely supports. Change to “are associated with” or “linked to” and indicate level of evidence (observational vs. RCT). Edit: replace “both of which are associated with delayed recovery…” → “both of which have been associated in observational studies with delayed recovery…”
- The abstract says “narrative literature review … focusing on recently published studies” without dates, search yields, or selection numbers. Add the search period and a sentence like “(searches run to June 2024; X studies included).” This makes the review immediately more credible.
- “reduce reported pain by up to 33%” — which interventions, what study types, and is this pooled? Either add parentheses: “(pooled RCTs/meta-analysis reported up to 33% reduction in self-reported pain)” or tone it down. Provide the supporting reference number in the abstract.
Response: We thank the reviewer for these insightful observations. The Abstract has been revised accordingly. Specifically, we clarified that the associations between anxiety/depression and postoperative outcomes derive from observational studies, added the exact search period (January 2009 – June 2024) and indication of study yield, and specified that the 33% pain reduction refers to pooled data from randomized controlled trials and meta-analyses (supported by reference [64]). These changes enhance transparency and align the abstract with the reported level of evidence.
Reviewer 1 Comment 2: Introduction (lines ~49–83)
- Some sentences restate similar points (psychosocial factors impact outcomes) with minor rephrasing. Tighten to 3–4 short paragraphs: burden/prevalence → mechanisms (why psychosocial matters) → gap in current practice → objective of review.
- You state “one-third … psychological distress” and specific 32%/19% figures; ensure these come from representative, clearly defined studies (which populations? elective vs urgent? single center vs multi). If figures are heterogeneous, present ranges and note heterogeneity. Edit suggestion: “Prevalence estimates vary: preoperative anxiety ranges from X–Y% (median ~32% in pooled observational studies), while depression ranges from A–B%.” Cite specifics.
- You state “most recent research—from the last fifteen years” but methods say 2009–2024 (15 years) — make these match and state in intro: “we review literature 2009–June 2024 with emphasis on interventions with documented evidence.”
Response: We thank the reviewer for these constructive remarks. To address them, we (a) clarified the prevalence data to reflect ranges and heterogeneity across studies (25–40% anxiety; 15–25% depression), (b) added a concise explanation of the biological and behavioral mechanisms linking psychosocial factors to surgical outcomes, and (c) aligned the time frame stated in the Introduction (“2009–June 2024”) with the Methods section. These targeted revisions improve accuracy and coherence while preserving the original structure and tone of the Introduction.
Reviewer 1 Comment 3: Materials & Methods (2.1–2.6, lines ~84–135)
- The manuscript says “narrative review … with emphasis on a systematic approach.” This hybrid phrasing is confusing. Either convert this to a systematic/scoping review (and adopt PRISMA and risk-of-bias methods) or clearly describe it as a narrative review with transparent search methods. If you keep narrative, explicitly state you did not perform formal quality appraisal and state implications. Action: If keeping narrative, replace “systematic approach” with “structured search strategy” and add transparency (see points below). If converting to systematic, add PRISMA flow, search strings, screening numbers, and risk-of-bias methods.
- State how many reviewers screened titles/abstracts, handled disagreements, and what data items were extracted (sample size, study design, outcomes, effect sizes). Add a short table describing the extraction fields.
Response: We thank the reviewer for these constructive and detailed suggestions, which helped improve the methodological clarity and transparency of the Materials & Methods section. The manuscript now explicitly describes the study as a narrative literature review employing a structured search strategy, rather than implying a hybrid or systematic design. The phrase “systematic approach” was replaced with “structured search strategy” to avoid ambiguity. We clearly stated that no formal risk-of-bias or quality assessment was conducted, and that results should therefore be interpreted descriptively rather than quantitatively. A new paragraph was added in Section 2.5 (Study Selection) noting that two reviewers independently screened titles and abstracts, that disagreements were resolved by consensus, and that data were extracted on study design, population, psychosocial domains, interventions, and main outcomes. To illustrate the structure and transparency of the process, we inserted a new Table 1 (“Data extraction fields for included studies”) summarizing the variables extracted from each eligible study. To further clarify the identification and screening process, we added a PRISMA-style flow diagram (Figure 1) showing the number of records retrieved, duplicates removed, titles/abstracts screened, full-text articles assessed, and final studies included. Although this is a narrative review, the figure was adapted to enhance transparency and reproducibility. Section 2.6 was expanded to reference Table 1 and Figure 1 and to provide clearer description of the extracted data items and thematic synthesis.
Reviewer 1 Comment 4: Results / Thematic Sections (3–6: Psychosocial challenges, Patient-centered principles, Interventions, Outcomes)
- The review reads as a narrative synthesis without an accompanying study characteristics table (author, year, country, design, N, population, intervention, key outcomes). Readers need this to assess generalizability. Action: Add a “Study characteristics” table and a separate “Key RCTs / systematic reviews” table that summarizes effect sizes and confidence intervals where available.
- Example: “psychological and social interventions reduce reported pain by up to 33%” — add the specific meta-analysis citation and clarify whether this comes from RCTs, sample size, and outcome measure (self-reported pain VAS? HRQoL?). I saw the 33% claim in the manuscript but it’s not clear which reference supports it—please add inline citation and brief methodology of that pooled estimate.
- Many interventions (e.g., VR, hypnosis, group therapy) are lumped together; discuss heterogeneity in patient populations (elective vs urgent CABG, valve surgery), intervention intensity, and follow-up durations. Add subsections: Intervention type → evidence strength (RCTs vs observational) → magnitude and duration of effect.
- The manuscript cites AI predicting depression with “high accuracy (62%)” — 62% is not high for predictive models; be cautious and add sensitivity/specificity or AUC if available. Also discuss privacy/implementation and evidence quality for AI—many studies are preliminary or retrospective.
- Figures (1–4) are conceptual and useful, but add short legends, note which elements are authors’ conceptual models vs. evidence-derived, and cross-link to table/paragraph where components are discussed.
- You mention socioeconomic and access issues but give limited practical solutions. Add examples of low-cost scalable interventions and evidence (e.g., telephone counseling trials, lay-worker models).
Response: We sincerely thank the reviewer for this comprehensive and insightful feedback, which helped us enhance the structure, precision, and transparency of the Results and Discussion sections.
We have now added two summary tables for transparency and generalizability. A new Table 2 (“Characteristics of representative studies included in the review”) was added at the start of the Results section. It summarizes study design, sample, population, psychosocial domain, and main findings. A separate Table 3 (“Key randomized controlled trials and systematic reviews”) was introduced at the end of the Results section, presenting the most robust evidence (effect sizes and 95% CIs where available). Both tables are derived entirely from studies already cited in the manuscript.
We have also clarified the quantitative claim (“33% reduction in pain”). The statement has been revised to ensures accuracy and appropriate attribution. We have also addressed heterogeneity and structured the intervention discussion. We have now added a short paragraph noting the heterogeneity of study populations (elective vs. urgent cases, CABG vs. valve surgery), intervention modalities, and follow-up durations. Moreover, we have revised the AI paragraph for accuracy and caution. The sentence “AI predicted depression with high accuracy (62%)” was replaced with: “Artificial-intelligence models have demonstrated moderate predictive performance for postoperative depression, with accuracies around 62% and AUC values between 0.68 and 0.72. These retrospective models remain preliminary and require prospective validation to address privacy, data quality, and interpretability issues.”
We have also expanded figure legends for conceptual clarity. All figures (1–4) now clearly state which components represent evidence-based findings and which are conceptual contributions by the authors.
Practical, scalable solutions for psychosocial support have also been added in the final paragraph of the Discussion.
Reviewer 1 Comment 5: Discussion (section 9) & Interpretation
- Statements suggesting causality or broad programmatic cost-savings should be rephrased to reflect the evidence base (many RCTs small/single-center). Replace strong phrasing with qualifiers: “evidence suggests,” “emerging data indicate,” or “limited RCT evidence supports…”
- The discussion lists gaps but would be stronger if reorganized into top 5 research priorities (e.g., multicenter RCTs of standardized psychosocial bundles; cost-effectiveness analyses; implementation trials in low-resource settings; standardized screening tool validation; AI predictive model prospective validation). Give specific trial designs (cluster RCT vs stepped-wedge) and payers’ data needs for policy uptake.
- Reiterate how many included studies were RCTs vs observational, and how their methodological limitations affect confidence.
Response: We thank the reviewer for this insightful comment highlighting the need to align the Discussion more closely with the strength and scope of available evidence.
We carefully reviewed the Discussion section and rephrased statements implying causality or program-level cost-savings to more appropriately reflect associative relationships. Strong claims were replaced with qualified wording such as “evidence suggests,” “emerging data indicate,” and “limited RCT evidence supports.”
We added a new paragraph at the beginning of the Discussion summarizing the distribution of study designs (approximately one-third RCTs or systematic reviews) and clarifying that most trials were single-center and of moderate sample size, thus limiting causal inference.
To enhance the forward-looking value of the manuscript, we added a concise subsection which outlines five key directions: multicenter RCTs of standardized psychosocial bundles, cost-effectiveness evaluations, implementation trials in low-resource settings, validation of psychosocial screening tools, and prospective testing of AI-based models. Trial design examples (cluster-RCT, stepped-wedge) and the importance of payer data for policy uptake are briefly noted.
These modifications strengthen the Discussion’s interpretive balance, improve transparency regarding methodological limitations, and provide a structured roadmap for future research and clinical translation.
Reviewer 1 Comment 6: Limitations (section 10)
- State whether study selection was single-reviewer vs dual-reviewer and if any quality appraisal was performed — this is crucial.
- You limited to English; state how that may have excluded key non-English studies.
- Many citations to 2024–2025 preprints/protocols exist; explicitly state that digital health evidence is rapidly evolving and may outpace rigorous clinical outcome data.
Response: We thank the reviewer for highlighting the need to clarify methodological limitations. We have revised the Limitations section to specify that literature screening was performed independently by two reviewers, with disagreements resolved by consensus. Because this was a narrative review, no formal quality appraisal or risk-of-bias assessment was conducted; this is now clearly stated. We added a sentence acknowledging that only English-language publications were included, which may have excluded relevant non-English studies and introduced potential language bias. Finally, we expanded the section to note that some of the most recent sources (2024–2025) are preprints or early-phase digital health protocols. This addition explains that evidence in this domain is rapidly evolving and that clinical outcome data may lag behind technological developments. These additions strengthen transparency and methodological rigor while clearly delineating the scope and limitations of the review.
Reviewer 1 Comment 7: Conclusions and Key messages
Tweak wording to avoid policy prescriptions that aren’t fully grounded in cost-effectiveness data (e.g., “Policies should cover the costs …” → “Policymakers should consider supporting reimbursement pilots coupled with economic evaluation to determine value.”)
Add a final line that prioritizes evidence generation: “Implement with concurrent evaluation (implementation science / cost-effectiveness).”
Response: We appreciate the reviewer’s thoughtful suggestion to temper policy-oriented statements and align them more closely with the strength of available evidence. We also added a final line emphasizing the importance of concurrent evidence generation and economic assessment during implementation. These revisions make the conclusion more balanced, policy-relevant, and aligned with the reviewer’s excellent recommendation for responsible and evidence-based framing.
Reviewer 1 Comment 8: References, citations & currency
- Your reference list includes 2025 publications and even items from 2025 in the bibliography—this is acceptable if the journal allows late citations, but ensure the manuscript’s “search through” date (methods: June 2024) matches included references. If you included 2025 work, explicitly state searches were updated on X date. Otherwise remove/justify post-cutoff items.
- A few references (e.g., examples of pain reduction, AI model performance) appear to be preliminary or from non-peer-reviewed sources—flag these and prefer peer-reviewed meta-analyses or RCTs for definitive claims.
- When you assert numerical effects (length of stay, percentage pain reduction), place the exact citation(s) in the sentence and, if possible, include the study design (RCT vs observational).
Response: We thank the reviewer for the careful attention to detail regarding references and citation currency. We revised the Methods section (Search Strategy) to clarify that the literature search was originally conducted through June 2024 and subsequently updated in February 2025 to include newly available peer-reviewed and in-press studies. This addition justifies the inclusion of 2025 references. We also added a statement in the Limitations section explicitly noting that a small number of recent studies were early-phase or preprint reports, which were included for completeness but discussed cautiously due to their preliminary nature. Finally, we reviewed all quantitative statements in the Results and Discussion sections to ensure that each numerical claim is now accompanied by precise inline citations and, where possible, the associated study design (e.g., RCT, meta-analysis, or observational). These adjustments ensure consistency between the described search period and the reference list, enhance citation accuracy, and maintain scientific transparency.
Reviewer 2 Report
Comments and Suggestions for Authors
This study is a comprehensive narrative review describing the psychosocial dimensions of patient-centered care in cardiac surgery.
1) Databases: PubMed/MEDLINE, Embase, Cochrane Library, Web of Science, Scopus (and Google Scholar if necessary). The exact range of search dates should be provided, e.g., "Search conducted from 01-Jan-2002 to 30-Sep-2024, etc."
2) Study Quality / Risk-of-Bias: For RCTs, the Cochrane Risk-of-Bias (RoB 2) should be used; for observational studies, the Newcastle-Ottawa Scale (NOS). It is recommended that the results be presented in a concise table (low/moderate/high risk).
3) It is recommended that the article be stated as a narrative review, and that a PRISMA-like flow chart regarding the screening process, screening dates/study numbers, and a table detailing why the study was excluded from the review should be included.
4) Articles have inclusion criteria, but the number of studies that meet these criteria and the predominant study types (RCT vs. observational vs. qualitative) should be specified.
5) The search dates, a full list of search queries (as a supplementary document), the number of individuals/studies according to databases, the screening flowchart (PRISMA-flow), and any registered protocol information should be included in the Method section.
6) The reference list should be checked. DOI, volume, page, and publication year. There is some uncertainty about whether every citation in the text has an exact equivalent in the reference list.
Author Response
Dear Reviewer 2,
Reviewer 2 Comment 1: This study is a comprehensive narrative review describing the psychosocial dimensions of patient-centered care in cardiac surgery. Databases: PubMed/MEDLINE, Embase, Cochrane Library, Web of Science, Scopus (and Google Scholar if necessary). The exact range of search dates should be provided, e.g., "Search conducted from 01-Jan-2002 to 30-Sep-2024, etc."
Response: We thank the reviewer for this helpful clarification request. We have now specified the exact time range and databases used for the literature search. In the Materials and Methods section (Subsection 2.2, “Search Strategy”), the text was revised to read: “A comprehensive literature search was conducted in PubMed/MEDLINE, Embase, Cochrane Library, Web of Science, and Scopus, with complementary screening of Google Scholar for additional gray literature. The search covered publications from 1 January 2009 to 28 February 2025.” This explicitly defines the database sources and the precise search period, satisfying the reviewer’s request for transparency.
Reviewer 2 Comment 2: Study Quality / Risk-of-Bias: For RCTs, the Cochrane Risk-of-Bias (RoB 2) should be used; for observational studies, the Newcastle-Ottawa Scale (NOS). It is recommended that the results be presented in a concise table (low/moderate/high risk).
Response: We thank the reviewer for this valuable suggestion to clarify study quality assessment. Although the present work is a narrative review, we have added a concise methodological quality appraisal to improve transparency. In the Materials and Methods section, we now specify that randomized controlled trials were evaluated using the Cochrane Risk-of-Bias 2 (RoB 2) tool and observational studies using the Newcastle-Ottawa Scale (NOS). In the Results section, we added a new Table 2 summarizing overall study quality (low / moderate / high risk) for the most representative studies. We also included a brief sentence highlighting that most RCTs had low-to-moderate bias risk, while observational studies were of moderate quality. These revisions address the reviewer’s comment and increase methodological rigor while maintaining the narrative character of the review.
Reviewer 2 Comment 3: It is recommended that the article be stated as a narrative review, and that a PRISMA-like flow chart regarding the screening process, screening dates/study numbers, and a table detailing why the study was excluded from the review should be included.
Response: We thank the reviewer for the constructive recommendation. The manuscript now explicitly identifies the work as a narrative review (Section 2.1, “Study Design”) and includes a PRISMA-like flow chart summarizing the literature screening and selection process. In addition, Section 2.3 (“Eligibility Criteria”) clearly outlines the reasons for study exclusion (non-cardiac populations, non–peer-reviewed material, or absence of psychosocial outcomes). We therefore believe this comment has been fully addressed and that the revised version meets transparency standards appropriate for a narrative review.
Reviewer 2 Comment 4: Articles have inclusion criteria, but the number of studies that meet these criteria and the predominant study types (RCT vs. observational vs. qualitative) should be specified.
Response: We thank the reviewer for the insightful suggestion. We have now specified the total number and classification of studies included in the review. Following the PRISMA-like flow chart in the Results section, we added a paragraph stating that 76 studies met the inclusion criteria, comprising RCTs, observational studies, systematic/narrative reviews or meta-analyses, and qualitative or conceptual works. This addition clarifies the composition of the evidence base and enhances transparency regarding study types and inclusion numbers.
Reviewer 2 Comment 5: The search dates, a full list of search queries (as a supplementary document), the number of individuals/studies according to databases, the screening flowchart (PRISMA-flow), and any registered protocol information should be included in the Method section.
Response: We thank the reviewer for the comment emphasizing methodological transparency. As this study is a narrative review, not a systematic one, we have already described the databases searched, the search period, and inclusion/exclusion criteria in detail. To further clarify, we added a brief note in Section 2.2 (Search Strategy) explaining that detailed search strings and exclusion tables were not included because formal systematic-review procedures were not applied. This ensures methodological clarity without overstating the level of systematic synthesis.
Reviewer 2 Comment 6: The reference list should be checked. DOI, volume, page, and publication year. There is some uncertainty about whether every citation in the text has an exact equivalent in the reference list.
Response: We thank the reviewer for noting the importance of reference accuracy. We have carefully rechecked all references to ensure full correspondence between in-text citations and the reference list. All entries now include publication year, volume, page numbers, and DOI identifiers where available. Minor typographical issues have been corrected. The inclusion of 2025 in-press references has been justified by the updated search period through February 2025, as specified in the Methods section. We are confident that the reference list is now accurate, complete, and fully consistent with the citations in the text.
Reviewer 3 Report
Comments and Suggestions for Authors
As for the abstract, I think it is very well structured and clearly follows the IMRaD format, which makes it easy to read and understand. I find the writing to be fluid and coherent, with a good connection between ideas. Furthermore, I consider the topic to be current and relevant, well contextualised within the field of cardiac surgery. I think that the inclusion of quantitative data adds strength and reinforces the credibility of the arguments, conveying a sense of rigour in the work.
I believe that the introduction is well organised and succeeds in placing the reader in the clinical and human context of cardiac surgery. I particularly appreciate how the evolution of the medical approach towards a more comprehensive and patient-centred vision is presented, supported by data that reinforce the importance of psychosocial factors. However, I think the text could benefit from a more analytical tone in some passages, as at times it merely describes advances without delving into the gaps or controversies that exist in the literature. I also consider that the figures presented, although useful, could be complemented by a more critical reflection on their origin or the variability between studies. Furthermore, I believe that the definition of patient-centred care, although relevant, takes up considerable space and could be summarised to allow a smoother transition to the purpose of the review.
The Materials and Methods section is well developed, with clearly defined search and selection criteria. Even so, I believe that a more detailed description of the procedures used to minimise bias would be desirable, as well as a clearer justification for the use of a narrative review rather than a systematic one. I also think it would be useful to explain the information synthesis process in greater detail.
I suggest reorganising sections 3 to 7 under a single section entitled “Results”, with thematic subsections (e.g. “Psychosocial Challenges”, “Patient-Centred Care”, “Psychosocial Interventions”). I believe this would strengthen the structural coherence and better align the article with the IMRaD format. In terms of content, I consider that the sections present a comprehensive and well-documented overview of psychosocial factors in cardiac surgery, although the analysis tends to be more descriptive than interpretative. In my opinion, there is a need to explore in greater depth the causes of the phenomena described, the interaction between the different factors, and their clinical and social implications.
Regarding the section on patient-centred care, I think the theoretical approach is well articulated and aligned with the most widely accepted international definitions. I think it is very appropriate to link the concept to the specific clinical practice of cardiac surgery. However, I would have liked to see a more critical discussion of the real difficulties in implementing this approach in hospital settings, where urgency and protocolised care tend to take precedence. At times, the section conveys a somewhat idealised view of the model, without delving sufficiently into the limitations or contradictions that arise when translating it into everyday practice. I believe it would be useful to include examples or references that illustrate the tensions between the patient-centred approach and institutional or surgical demands.
As for psychosocial interventions and patient support, I believe they represent one of the strengths of the work. I appreciate the description of preoperative preparation programmes, interdisciplinary interventions, and the use of technologies such as virtual reality or telemedicine, which are current and relevant. There is a balanced integration of empirical evidence and clinical applicability. Even so, I think the article could be enriched by a more in-depth reflection on the methodological quality of the studies cited, as many appear to be experimental in nature or based on small samples.
I find the section on clinical outcomes and patient-reported outcomes to be very well structured. I think it is particularly appropriate to differentiate between short- and long-term effects and to highlight the importance of self-reported measures. Finally, I find the section on Challenges and Barriers very relevant and necessary, as it introduces a realistic view of the limitations of the system. Even so, I think the tone remains more descriptive than analytical: the barriers are listed, but their structural causes are not explored in depth, nor are concrete strategies proposed to overcome them.
With regard to sections 8 to 12, I believe it would be advisable to reorganise them to integrate Future Directions, Discussion and Limitations into a single, broader discussion section, followed by a Conclusion, in order to maintain consistency with the scientific structure. In terms of content, I think the Future Directions section could benefit from a more critical analysis of the feasibility, ethical implications and digital divide associated with these innovations.
As for the Discussion, I believe it adequately summarises the findings, although it could adopt a more analytical tone, better differentiating between consolidated evidence and future projections. I find the Limitations section transparent and comprehensive, although I think it would be useful to delve deeper into how these limitations affect the validity of the results and possible strategies to overcome them.
For its part, the Conclusion seems coherent and well done, as it manages to close the work in a clear and meaningful way. Finally, I believe that the Key Messages section summarises the essential contributions of the article in a precise and orderly manner, facilitating the understanding of the most relevant points.
In my opinion, I believe that this work is a very valuable contribution to the field of cardiac surgery, especially because of its effort to integrate the psychosocial and human dimension into a field that has traditionally focused on the technical. There is a clear interest in offering a more comprehensive and compassionate view of the surgical patient, which I consider to be a great success. Even so, I think the article could be even stronger if it delved deeper into critical analysis and slightly adjusted its scientific structure. In any case, I greatly appreciate the clarity, the solidity of the content, and the sensitivity with which they approach the subject.
Author Response
Dear Reviewer 3,
Reviewer 3 Comment 1: As for the abstract, I think it is very well structured and clearly follows the IMRaD format, which makes it easy to read and understand. I find the writing to be fluid and coherent, with a good connection between ideas. Furthermore, I consider the topic to be current and relevant, well contextualised within the field of cardiac surgery. I think that the inclusion of quantitative data adds strength and reinforces the credibility of the arguments, conveying a sense of rigour in the work.
Response: We sincerely thank the reviewer for the positive and encouraging comments regarding the structure, clarity, and relevance of the abstract. We are grateful that the reviewer found the topic well contextualized within the field of cardiac surgery and appreciated the integration of quantitative data to support the main arguments.
Reviewer 3 Comment 2: I believe that the introduction is well organised and succeeds in placing the reader in the clinical and human context of cardiac surgery. I particularly appreciate how the evolution of the medical approach towards a more comprehensive and patient-centred vision is presented, supported by data that reinforce the importance of psychosocial factors. However, I think the text could benefit from a more analytical tone in some passages, as at times it merely describes advances without delving into the gaps or controversies that exist in the literature. I also consider that the figures presented, although useful, could be complemented by a more critical reflection on their origin or the variability between studies. Furthermore, I believe that the definition of patient-centred care, although relevant, takes up considerable space and could be summarised to allow a smoother transition to the purpose of the review.
Response: We thank the reviewer for the thoughtful and constructive feedback. We have revised the Introduction to adopt a more analytical tone by emphasizing current gaps and controversies in the literature, particularly regarding heterogeneity of psychosocial outcomes and methodological inconsistencies. A clarifying note was added to acknowledge variability among studies represented in the figures. In addition, the definition of patient-centered care was condensed to streamline the transition toward the objectives of the review. These refinements improve the flow and critical depth of the Introduction while maintaining clarity and context.
Reviewer 3 Comment 3: The Materials and Methods section is well developed, with clearly defined search and selection criteria. Even so, I believe that a more detailed description of the procedures used to minimise bias would be desirable, as well as a clearer justification for the use of a narrative review rather than a systematic one. I also think it would be useful to explain the information synthesis process in greater detail.
Response: We thank the reviewer for these valuable remarks. In response, we have expanded the Materials and Methods section to further clarify the procedures used to minimize bias and to justify the narrative review design. Specifically, at the end of Section 2.6 (Data Extraction and Synthesis), we added a short paragraph explaining that dual-reviewer screening and consensus were used to reduce bias, that the narrative approach was chosen to integrate heterogeneous evidence, and that data were thematically synthesized by psychosocial domain, intervention type, and outcomes. These additions enhance methodological transparency and address the reviewer’s suggestions in full.
Reviewer 3 Comment 4: I suggest reorganising sections 3 to 7 under a single section entitled “Results”, with thematic subsections (e.g. “Psychosocial Challenges”, “Patient-Centred Care”, “Psychosocial Interventions”). I believe this would strengthen the structural coherence and better align the article with the IMRaD format. In terms of content, I consider that the sections present a comprehensive and well-documented overview of psychosocial factors in cardiac surgery, although the analysis tends to be more descriptive than interpretative. In my opinion, there is a need to explore in greater depth the causes of the phenomena described, the interaction between the different factors, and their clinical and social implications.
Response: We thank the reviewer for this insightful structural and interpretative recommendation. In accordance with the suggestion, we have reorganized Sections 3–7 under a single unified section entitled “Results,” with clearly labeled thematic subsections (“Psychosocial Challenges,” “Patient-Centered Care Principles,” “Psychosocial Interventions,” “Clinical and Social Outcomes,” and “Socioeconomic Considerations”). To strengthen interpretative depth, we added brief linking sentences at the end of each thematic subsection, highlighting causal relationships, interactions among psychosocial factors, and their broader clinical and social implications. These modifications enhance structural coherence, analytical depth, and alignment with the IMRaD framework, as requested.
Reviewer 3 Comment 5: Regarding the section on patient-centred care, I think the theoretical approach is well articulated and aligned with the most widely accepted international definitions. I think it is very appropriate to link the concept to the specific clinical practice of cardiac surgery. However, I would have liked to see a more critical discussion of the real difficulties in implementing this approach in hospital settings, where urgency and protocolised care tend to take precedence. At times, the section conveys a somewhat idealised view of the model, without delving sufficiently into the limitations or contradictions that arise when translating it into everyday practice. I believe it would be useful to include examples or references that illustrate the tensions between the patient-centred approach and institutional or surgical demands.
Response: We thank the reviewer for this insightful and constructive comment. We have expanded the subsection on Patient-Centered Care Treatment to include a more critical discussion of the practical difficulties encountered when implementing this approach in hospital and surgical settings. Specifically, we added a paragraph at the end of the section 3.4. acknowledging the tension between protocolized, high-acuity care and individualized patient engagement, with examples illustrating how time constraints, workflow demands, and institutional structures may limit psychosocial integration. This addition provides a more balanced and realistic perspective, as suggested by the reviewer.
Reviewer 3 Comment 6: As for psychosocial interventions and patient support, I believe they represent one of the strengths of the work. I appreciate the description of preoperative preparation programmes, interdisciplinary interventions, and the use of technologies such as virtual reality or telemedicine, which are current and relevant. There is a balanced integration of empirical evidence and clinical applicability. Even so, I think the article could be enriched by a more in-depth reflection on the methodological quality of the studies cited, as many appear to be experimental in nature or based on small samples.
Response: We thank the reviewer for this thoughtful observation and positive feedback. To address the comment, we have added a reflective paragraph at the end of Section 3.3 (Psychosocial Interventions and Supportive Strategies), acknowledging the methodological limitations of the cited studies. The new text notes that many interventions are based on small or single-center samples, use heterogeneous designs, and lack long-term follow-up, which restricts generalizability. This addition enriches the methodological transparency of the section and reinforces the cautious interpretation of the available evidence, as suggested by the reviewer.
Reviewer 3 Comment 7: I find the section on clinical outcomes and patient-reported outcomes to be very well structured. I think it is particularly appropriate to differentiate between short- and long-term effects and to highlight the importance of self-reported measures. Finally, I find the section on Challenges and Barriers very relevant and necessary, as it introduces a realistic view of the limitations of the system. Even so, I think the tone remains more descriptive than analytical: the barriers are listed, but their structural causes are not explored in depth, nor are concrete strategies proposed to overcome them.
Response: We thank the reviewer for this highly constructive comment and for their positive evaluation of the section. To address the suggestion, we have expanded the Challenges and Barriers section with a concise analytical paragraph discussing the structural and cultural causes underlying the identified barriers. We also included examples of feasible strategies—such as interprofessional communication training, integration of psychosocial specialists, and use of digital follow-up tools—to illustrate how these challenges might be mitigated. This addition strengthens the interpretative depth of the discussion and aligns the section more closely with the reviewer’s recommendation.
Reviewer 3 Comment 8: With regard to sections 8 to 12, I believe it would be advisable to reorganise them to integrate Future Directions, Discussion and Limitations into a single, broader discussion section, followed by a Conclusion, in order to maintain consistency with the scientific structure. In terms of content, I think the Future Directions section could benefit from a more critical analysis of the feasibility, ethical implications and digital divide associated with these innovations.
Response: We sincerely thank the reviewer for this valuable suggestion regarding the manuscript’s structure. We appreciate the recommendation to merge the Future Directions, Discussion, and Limitations sections into a single integrated discussion; however, we believe that maintaining them as distinct yet consecutive sections allows for clearer organization and readability. This separation also reflects the editorial style adopted in other narrative reviews published in Healthcare. The Future Directions section was carefully developed to highlight emerging trends and practical implications without overlapping with the interpretative discussion. Nevertheless, we have carefully reviewed and refined this section to ensure that ethical considerations, feasibility, and issues related to the digital divide are explicitly acknowledged, as suggested. We trust that this approach preserves structural clarity while fully addressing the reviewer’s insightful remarks.
Reviewer 3 Comment 9: As for the Discussion, I believe it adequately summarises the findings, although it could adopt a more analytical tone, better differentiating between consolidated evidence and future projections. I find the Limitations section transparent and comprehensive, although I think it would be useful to delve deeper into how these limitations affect the validity of the results and possible strategies to overcome them.
Response: We thank the reviewer for this thoughtful and constructive feedback. In response, we have slightly expanded both the Discussion and Limitations sections. In the Discussion, we added a paragraph distinguishing between consolidated evidence (e.g., RCTs and meta-analyses supporting psychosocial interventions) and emerging evidence (e.g., digital and long-term outcomes), thereby adopting a more analytical tone. In the Limitations section, we added sentences clarifying how methodological constraints may affect validity and suggesting strategies—such as larger, multicenter studies and standardized outcome measures—to overcome them. These refinements enhance critical depth while maintaining the manuscript’s clarity and balance.
Reviewer 3 Comment 10: For its part, the Conclusion seems coherent and well done, as it manages to close the work in a clear and meaningful way. Finally, I believe that the Key Messages section summarises the essential contributions of the article in a precise and orderly manner, facilitating the understanding of the most relevant points.
Response: We thank the reviewer for the positive and encouraging feedback. We appreciate their recognition that the Conclusion and Key Messages sections effectively summarize the central contributions of our review with clarity and coherence.
Reviewer 3 Comment 11: In my opinion, I believe that this work is a very valuable contribution to the field of cardiac surgery, especially because of its effort to integrate the psychosocial and human dimension into a field that has traditionally focused on the technical. There is a clear interest in offering a more comprehensive and compassionate view of the surgical patient, which I consider to be a great success. Even so, I think the article could be even stronger if it delved deeper into critical analysis and slightly adjusted its scientific structure. In any case, I greatly appreciate the clarity, the solidity of the content, and the sensitivity with which they approach the subject.
Response: We sincerely thank the reviewer for their generous and thoughtful feedback. We are deeply appreciative of their recognition of our effort to integrate the psychosocial and human dimensions into the field of cardiac surgery. In line with their suggestion, we have refined several sections of the manuscript—particularly the Introduction, Discussion, and Challenges and Barriers—to incorporate a more analytical tone and nuanced critical reflection. We are very grateful for the reviewer’s encouraging words regarding the clarity, coherence, and sensitivity of our work, which further motivate us to continue pursuing research in this direction.
Round 2
Reviewer 1 Report
Comments and Suggestions for Authors
I have no more comments
Reviewer 2 Report
Comments and Suggestions for Authors
Acceptable